# Reconstitution of human DNA licensing and the structural and functional analysis of key intermediates

Jennifer N. Wells[1,2,3], Lucy V. Edwardes[1,2,3], Vera Leber[1,2], Shenaz Allyjaun[1,2], Matthew Peach [1,2], Joshua Tomkins[1,2], Antonia Kefala-Stavridi[1,2], Sarah V. Faull [1,2], Ricardo Aramayo[1,2], Carolina M. Pestana[1,2], Lepakshi Ranjha[1,2] & Christian Speck [1,2] ✉

Human DNA licensing initiates replication fork assembly and DNA replication. This reaction promotes the loading of the hMCM2-7 complex on DNA, which represents the core of the replicative helicase that unwinds DNA during S-phase. Here, we report the reconstitution of human DNA licensing using purified proteins. We showed that the in vitro reaction is specific and results in the assembly of high-salt resistant hMCM2-7 double-hexamers. With ATPγS, an hORC1-5-hCDC6-hCDT1-hMCM2-7 (hOCCM) assembles independent of hORC6, but hORC6 enhances double-hexamer formation. We determined the hOCCM structure, which showed that hORC-hCDC6 recruits hMCM2-7 via five hMCM winged-helix domains. The structure highlights how hORC1 activates the hCDC6 ATPase and uncovered an unexpected role for hCDC6 ATPase in complex disassembly. We identified that hCDC6 binding to hORC1-5 stabilises hORC2-DNA interactions and supports hMCM3-dependent recruitment of hMCM2-7. Finally, the structure allowed us to locate cancer-associated mutations at the hCDC6-hMCM3 interface, which showed specific helicase loading defects.

Prior to cell division, the entire genome needs to be duplicated. This process is split into multiple steps. It starts with loading the replicative helicase onto DNA in the late M-/early G1- phase, followed by activation of the helicase during the G1/S transition and, finally, assembly of the full replisome and DNA synthesis in S-phase[1,2]. Helicase loading, also termed pre-replication complex (pre-RC) assembly[3] and DNA licensing[4], is a multi-step process[5–7].

The reaction has been widely studied in budding yeast, where the yeast origin recognition complex 1-6 (yORC) binds to replication origins in a sequence-specific fashion[8,9]. In late M-phase, yCdc6 is recruited to yORC[10,11] and the yORC-yCdc6 complex encircles DNA[12,13]. Consequently, and with the help of yCdt1, a spiral-shaped hexameric yMCM2-7 becomes recruited[14,15], resulting in a yORC-yCdc6-yCdt1-yMCM2-7 (yOCCM) complex[16]. During yOCCM formation, the

C-terminal winged helix domains (WHD) of yMcm3 and yMcm7 make the first contact with yORC-yCdc6[16–18]. Then, a yMcm6-yCdt1 interaction promotes the insertion of the DNA into the yMCM2-7 helicase followed by yMCM2-7 closure around DNA[16,19] and the open-spiral to closed-ring yMCM2-7 hexamer transition. The induction of ATP-hydrolysis triggers the release of yCdc6 and yCdt1[5,7]. Subsequently, yORC becomes repositioned to the N-terminal face of the loaded yMCM2-7 hexamer[20], which promotes another round of yCdc6 and yCdt1-dependent loading of yMCM2-7[21,22], resulting in yMCM2-7 double-hexamer formation[23,24]. This large complex encircles double-stranded DNA and assumes an inactive state. During the G1/S transition, the helicase becomes activated and competent for DNA synthesis[23,24]. During helicase activation, the yMCM2-7 double-hexamer splits and becomes integrated into two replication forks, which

[1]DNA Replication Group, Institute of Clinical Sciences, Faculty of Medicine, Imperial College London, London, UK. [2]MRC Laboratory of Medical Sciences (LMS), London, UK. [3]These authors contributed equally: Jennifer N. Wells, Lucy V. Edwardes. ✉e-mail: chris.speck@imperial.ac.uk

replicate DNA bi-directionally[1,2,6]. yMCM2-7 is the replicative helicase's motor element within the replication fork, consisting of yCdc45, yMCM2-7 and yGINS (yCMG)[25].

The reconstitution of budding yeast DNA licensing[23,24] was instrumental in understanding the process of helicase loading, its regulation and the discovery of the protein structures of the respective intermediates[7,18–20]. By comparison, human DNA licensing has yet to be reconstituted and is therefore understudied, although the human ORC (hORC) complex has been analysed both structurally and biochemically[26–31]. This analysis revealed that hORC1-5 forms a C-shaped complex similar to yORC. hORC1, hORC4, hORC5 belong to the AAA+ family of ATPases and bind ATP at the subunit interfaces[27,31], while hORC2 and hORC3 adopt an AAA+ like organisation but do not bind ATP. Finally, hORC6 adopts a TFIIB-like structure that contains a DNA binding domain[32]. Compared to the yeast complex, hORC adopts a more flexible configuration[28–30]. In particular, hORC1 toggles between an ATPase inactive and active state, while the hORC2-WHD has been observed in two conformations, inactive and autoinhibited, which block hORC-DNA interactions[26,27,31]. Interestingly, hORC6 associates only weakly with hORC1-5[28–30,33] and its role in DNA replication remains ambiguous[32,34]. Whether the conformational flexibility of hORC has a functional or regulatory role in pre-RC formation is unknown.

hMCM2-7 forms a stable complex in solution that adopts a spiral configuration similar to yMCM2-7, but unlike the yeast complex, does not co-purify with hCDT1[23,35]. Thus, how hCDT1 becomes recruited during human DNA licensing is not entirely clear. The structure of the hMCM2-7 double-hexamer was recently revealed and identified that base-pairing interactions near the double-hexamer interface are destabilised[36], which may slow down the sliding of the complex on DNA under low salt conditions.

Although some human replication origins have been identified, it is clear that in contrast to yORC, hORC does not interact with DNA in a sequence-specific fashion. ORC from budding yeast contains an additional alpha helix in yOrc4, an essential contributor to its DNA sequence specificity that is missing in hORC4[16,37–39]. Instead, it is thought that the hORC1 N-terminal bromo-adjacent homology domain could be important in chromatin-mediated recruitment of metazoan ORC1-5[40,41].

The control of human DNA licensing is crucial. Licensing of DNA that has already been replicated leads to re-replication, promotes recombination and genomic instability[42]. Equally, too little DNA licensing will limit DNA synthesis[43]. In human cells, an excess of hMCM2-7 is loaded, which serves as dormant origins in case an active replication fork becomes terminally stalled[44–47]. However, not only is the frequency of DNA licensing important but so is its speed. Specifically, stem cells require rapid DNA licensing to maintain their pluripotency[48]. Therefore, the process of helicase loading is under the control of the DNA licensing checkpoint that monitors whether the genome has been sufficiently licensed for DNA replication. p53 and RB have an essential role in this checkpoint[49]. Interestingly, cancer cells frequently lose this checkpoint and display greater genomic instability[50]. Therefore, developing therapeutics targeting the DNA licensing machinery represents a promising research avenue[51].

To date, a number of cancer-associated mutations, as well as several mutations linked to the Meier-Gorlin syndrome, a rare human disease associated with primordial dwarfism[52], have been mapped to DNA licensing factors. Still, their functional relevance has yet to be discovered due to a lack of structural information and an efficient in vitro assay to test the impact of the mutations[53–55]. Here, we report the establishment of a fully reconstituted DNA licensing assay incorporating full-length (FL) proteins, which leads to hMCM2-7 double-hexamer formation. We show that hORC6, in contrast to yOrc6, is not essential for high-salt stable MCM2-7 loading but improves the efficiency of the reaction. Our data show that hORC6 is only recruited late during DNA licensing, suggesting that it functions in the loading of the second hMCM2-7 hexamer. Consistently, we observed increased hMCM2-7 double-hexamer formation in the presence of hORC6. By blocking ATP-hydrolysis, it is possible to assemble a hOCCM intermediate, which incorporates all human DNA licensing factors except hORC6. We obtained the structure of the hOCCM using cryo-EM and observed the hMCM2-7 encircling DNA, which had already been inserted into the complex. This structure provides key insight into hORC1-5-DNA interactions that allow us to rationalise how hCDC6 binding influences structural changes in hORC1-5 that are essential for hMCM2-7 recruitment. Finally, we used the structure to probe the impact of cancer-associated mutations on DNA licensing.

## Results and discussion

### hORC1-5 binding to DNA is stabilised by hCDC6

Helicase loading is a multi-step process that hORC-hCDC6 initiates[56]. We started off investigating the ability of hORC and hCDC6 to form a complex and bind to DNA. We focussed on FL hORC proteins, as truncations in *Drosophila* Orc1 have been linked to reduced DNA binding[57] and an N-terminal deletion of hORC2 has been linked to an altered cell cycle profile[58]. The hORC1-5 complex was overexpressed in budding yeast cells arrested in G1 phase, when S-phase specific cyclin-dependent kinases are inactive and co-purified as a pentameric complex (Fig. 1a, lane 1 and Supplementary Fig. 1a). For the purification, as well as for pull-down assay hORC1 contained an N-terminal Twin Strep-tag (hORC1 N-tag). As previous work suggested that hORC6 does not form a stable complex with hORC1-5[29], hORC6 was expressed in bacteria and purified as a monomer (Fig. 1a, lane 2 and Supplementary Fig. 1b). hCDC6 was similarly expressed in bacteria and purified as a homogenous monomer (Fig. 1a, lane 3 and Supplementary Fig. 1c). Next, we exploited the Strep-tag on hORC1 for use in a pulldown assay, in order to assess whether hORC1-5, hORC6 and hCDC6 would form a complex in vitro. We observed that all hORC1-5 subunits were retained in the hORC1 N-tag pulldown, though hORC6 and hCDC6 did not co-precipitate with hORC1-5 when compared to the non-specific control (Fig. 1b, compare lanes 2 and 3 with 4 and 5). By contrast, when the reaction was repeated in the presence of a 90 bp double-stranded (ds) DNA, hCDC6 was retained with hORC1-5 but not hORC6 (Fig. 1b, lanes 7 and 8). Taken together, these results indicate that neither hORC1-5-hCDC6 nor hORC1-5-hORC6 form a stable complex in solution and that the interaction between hORC1-5-hCDC6 is DNA dependent.

To further validate this observation, we performed an assay with a biotinylated 3 kbp human B2-lamin replication origin dsDNA sequence, coupled to streptavidin-coated magnetic beads. Proteins were incubated with DNA for 20 min at 30 °C in an ATP-containing low salt buffer to facilitate complex assembly on DNA. After the unbound protein was washed away, the DNA was cleaved by endonuclease DNaseI, which specifically releases DNA-bound proteins. In contrast, proteins non-specifically attached to the beads will not elute. hORC1-5 by itself associated specifically with dsDNA (Fig. 1c, lane 1). Notably, we observed significantly enhanced hORC1-5 binding upon inclusion of hCDC6 (Fig. 1c, lane 1–3 and Supplementary Fig. 2a). Moreover, hORC1-5 binding was observed across ten different 300 bp B2-lamin sub-fragments (Supplementary Fig. 2b), consistent with the concept that hORC1-5 has no specific sequence specificity. Taken together, these results indicate that hORC1-5 binds to DNA in a sequence-independent manner, that hORC6 is dispensable for the initial binding of hORC1-5-hCDC6 to DNA, and that hCDC6 enhances hORC1-5 binding to DNA.

### Reconstitution of human DNA licensing in vitro

Establishing an in vitro assay for human DNA licensing has the potential to reveal how the process works at a mechanistic level, enables structural studies and represents a platform to explore the role of patient-derived mutations in helicase loading. FL hMCM2-7 was expressed in Human Embryonic Kidney (HEK) 293F cells and purified as a hexamer (Fig. 1d and Supplementary Fig. 1d). FL hCDT1 and

N-terminally truncated hCDT1$_{\Delta N}$ (aa158-546) were expressed in *E. coli* and purified as monomers, though yields for hCDT1$_{\Delta N}$ were much higher (Fig. 1e and Supplementary Fig. 1e, f). Consequently, we tested hORC1-5, hORC6, hCDC6, hCDT1 or hCDT1$_{\Delta N}$ and hMCM2-7 at a relative ratio of 1: 2: 2: 1: 1.5 respectively (Fig. 1f, lanes 1–6) in the pre-RC assay using a 2 kbp double-stranded human Lamin B2 replication origin bound to magnetic beads. Here, proteins are assembled on DNA in the presence of ATP and either washed with low-salt buffer preserving reaction intermediates, or, with high-salt buffer revealing a salt-stable hMCM2-7 complex that is encircling DNA, which represents the final product of the reaction. After the wash, the DNA-bound proteins are eluted from the beads by DNaseI digest. In the low salt wash elution, we observed that all the DNA licensing factors, including hORC6, were in complex with DNA (Fig. 1f, lanes 7 and 8) and we detected a stable hMCM2-7 loading product in the high-salt washed sample (Fig. 1f, lanes 9 and 10). These results were consistent when testing with both FL and hCDT1$_{\Delta N}$ (Fig. 1f, compare lane 7–8 and 9–10). As yields from hCDT1$_{\Delta N}$ purifications were an order of magnitude higher, this construct was predominantly used for subsequent pre-RC assays.

## Human MCM2-7 forms a salt-stable double-hexamer

It has been observed that chromatin isolated from both yeast and human cells contains high salt-resistant MCM2-7 double-hexamers[59,60]. To understand the relative stability of yeast and human DNA licensing complexes, we assembled pre-RC reactions and then washed them with buffer containing increasing concentrations of sodium chloride. The yeast DNA licensing proteins, yORC, yCdc6, yCdt1 and yMCM2-7 clearly assembled into a complex, which was stable in low salt conditions (Fig. 2a, lane 2). However, after the addition of sodium chloride, yORC, yCdc6 and yCdt1 are readily released (Fig. 2a, lane 3). Also, a fraction of yMCM2-7 was retained, and it remained stable when washed with up to 500 mM sodium chloride (Fig. 2a, lanes 3–6). The human DNA licensing proteins were also observed to assemble into a complex on DNA (Fig. 2b, lane 2). Upon addition of lower concentrations of sodium chloride, some hORC was retained (Fig. 2b, lanes 3–4), while at higher salt concentrations, only hMCM2-7 was retained (Fig. 2b, lanes 5–6). Thus, budding yeast and human DNA licensing result in MCM2-7 complexes with nearly identical salt stability. This indicates that DNA licensing results in the topological entrapment of DNA and the

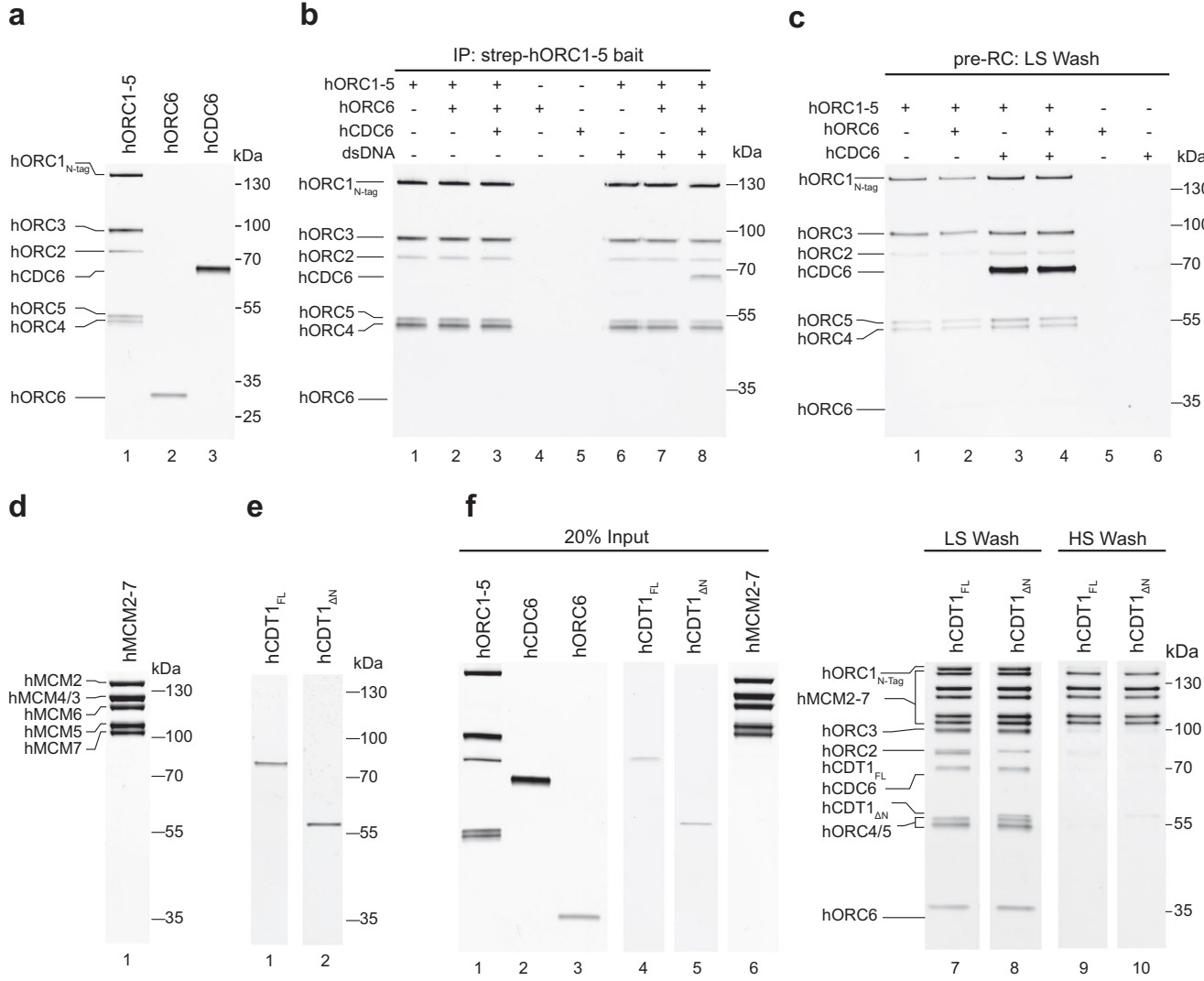

**Fig. 1 | Reconstitution of human DNA licensing using purified proteins.**
**a** Purified hORC1-5, hORC6 and hCDC6. **b** IP pull down with Strep-hORC1-5 as bait in the presence and absence of a 90 bp yeast ARS1 DNA containing the A, B1 and B2 elements. **c** Pre-RC assay performed under low salt conditions with hORC1-5, hCDC6 and hORC6 independently and in combination. **d** Purified hMCM2-7.

**e** Purified full-length hCDT1 and hCDT1$_{\Delta N}$ (aa158-546). **f** Pre-RC assays using hCDT1 and hCDT1$_{\Delta N}$, under low and high salt conditions in pre-RC buffer which contains 1 mM ATP. All SDS-PAGE gels are representative of three independent biological replicates. Source data are provided as a Source Data file.

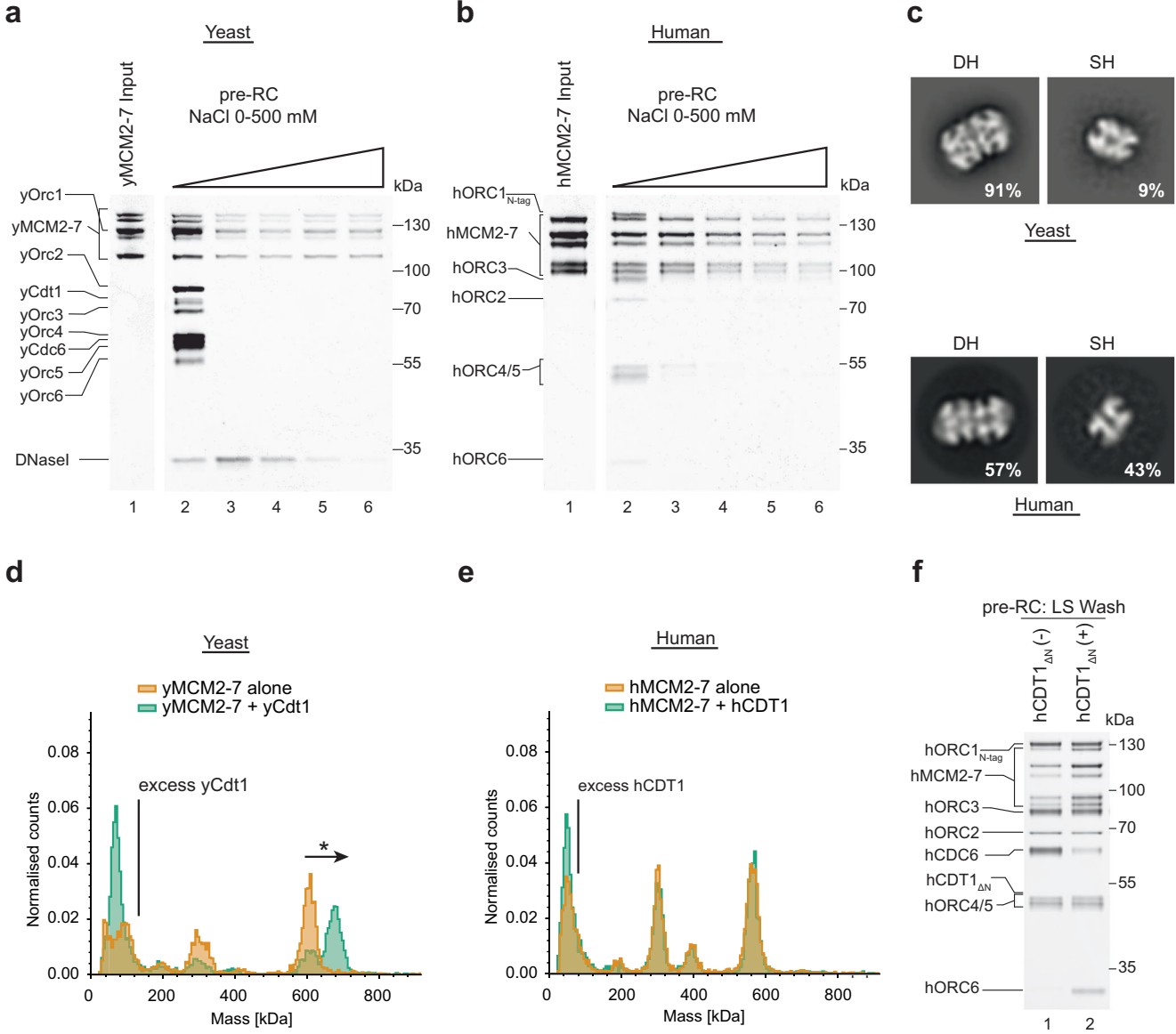

**Fig. 2 | Comparative analysis of double-hexamer formation and Cdt1 interaction with MCM2-7 using purified yeast and human proteins.** Assembled pre-RC reactions were subjected to washes with a sodium chloride gradient ranging from 0 to 500 mM for (**a**) yeast and (**b**) human. SDS-PAGE gels are representative of two independent biological replicates. **c** Negative stain EM 2D class averages generated with CryoSPARC carried out on pre-RC assay reactions washed with 300 mM NaCl from yeast and human. The percentage numbers represent the proportion of the total particles in each class of either double hexamer (DH) or single hexamer (SH). hMCM2-7 DH formation was more variable than yMcm2-7 DH-formation,

suggesting additional regulation mechanisms exist. Mass photometry on (**d**) yMCM2-7 and yCdt1 and (**e**) hMCM2-7 and hCDT1$_{\Delta N}$, was carried out in solution with a two-fold excess of CDT1. Only yeast proteins form a complex in solution, as detected by an upshift (marked with an asterisk) in the yMCM2-7 mass peak that is proportional to the mass of yCdt1. **f** Pre-RC assay under low salt conditions was carried out in the absence and presence of hCDT1$_{\Delta N}$, highlighting that initial hMCM2-7 recruitment occurs without hCDT1. SDS-PAGE gel is representative of three independent biological replicates. Source data are provided as a Source Data file.

establishment of hydrophobic protein-interactions that withstand the high-salt wash.

Next, we wanted to ask whether the final human helicase loading product assumes the form of the MCM2-7 double-hexamer, as previously observed with the budding yeast reaction[23,24,61]. Negative stain electron microscopy was applied to eluates from the high salt-washed pre-RC reactions carried out using both the human and yeast systems. The 2D class averages confirm the formation of hMCM2-7 double hexamer for yMCM2-7 (Fig. 2c) and for hMCM2-7 (Fig. 2c). Interestingly, in the context of human DNA licensing, a larger fraction of single-MCM2-7 was retained. This could be due to reduced hMCM2-7 double hexamer stability or could suggest that additional protein factors or specific DNA sequences may regulate the reaction. Future work has the

potential to reveal how human DNA licensing can be regulated by post-translational modifications or chaperones, which could change the reaction dynamics. In summary, this data shows that the human DNA licensing assay supports high salt-stable, hMCM2-7 complex formation on DNA and the assembly of hMCM2-7 double hexamers, representing the reaction's end product. Thus, the human DNA licensing assay is fully functional and well-suited for structural and functional analysis.

**The role of hCDT1 in hMCM2-7 recruitment**

In budding yeast, yMCM2-7 and yCdt1 form a stable complex in solution, with yCdt1 acting to maintain the spiral shape of yMCM2-7[14]. hMCM2-7 purified from human cells does not co-purify with hCDT1[35],

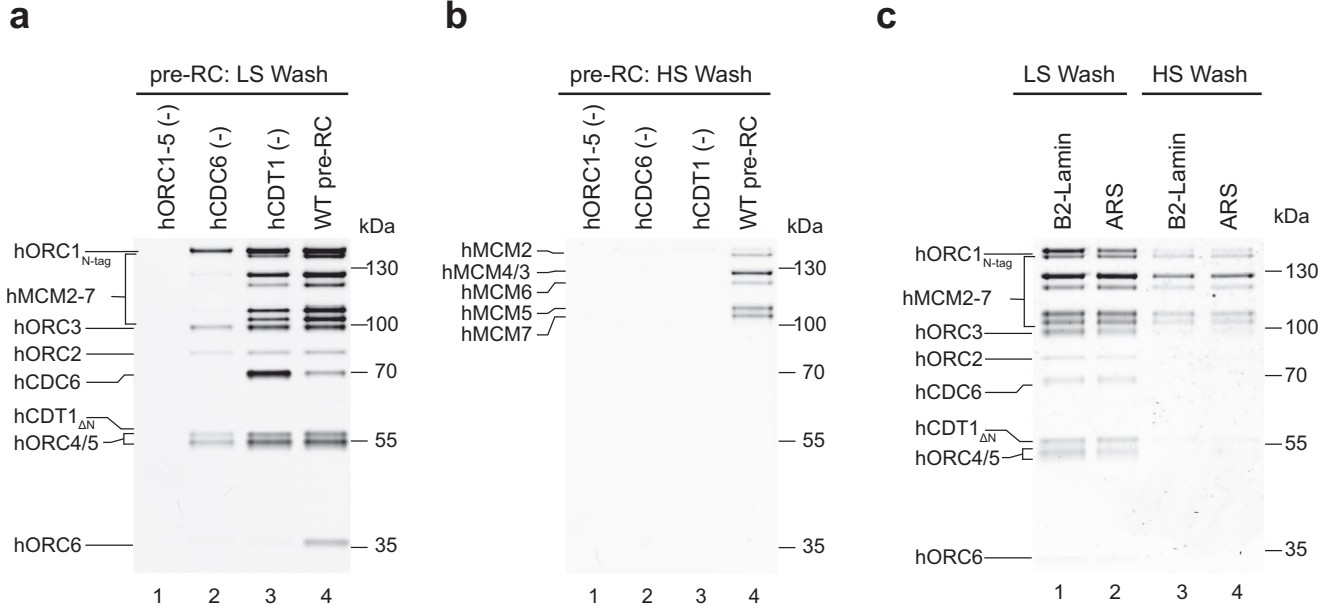

**Fig. 3 | Specificity of human DNA licensing.** Specificity of the pre-RC assay was assessed under (**a**) low and (**b**) high salt conditions by sequentially omitting loading factors hORC1-5 (lane 1), hCDC6 (lane 2) or hCDT1 (lane 3) compared to a reaction containing all pre-RC proteins (lane 4). SDS-PAGE gels are representative of three independent biological replicates. **c** Pre-RC assay under low and high salt wash conditions was carried out with hB2-Lamin and yARS1 DNA. Highlighting that human DNA licensing can occur on non-human DNA. SDS-PAGE gel is representative of two independent biological replicates. Source data are provided as a Source Data file.

however, hCDT1 was found to interact with hMCM6 in pulldown assays[62]. We used mass-photometry to analyse whether hCDT1$_{\Delta N}$ and hMCM2-7 interact in solution, using homologous yeast proteins as a positive control. We observed that the addition of yCdt1 to yMCM2-7 resulted in a yCdt1-yMCM2-7 complex, identifiable by an upshift in the maximum molecular mass detected (Fig. 2d), but this was not true for human homologues (Fig. 2e). The same results were obtained using hCDT1$_{FL}$ (Supplementary Fig. 2c). Even when challenged by a 10-fold excess of hCDT1$_{FL}$ relative to hMCM2-7, we did not observe complex formation (Supplementary Fig. 2d). This result raised the question as to whether hCDT1 is required for the initial binding of hMCM2-7. To address this question, we performed pre-RC assembly reactions in the absence and presence of hCDT1$_{\Delta N}$. When hCDT1$_{\Delta N}$ was omitted, we observed, to our surprise, an association of all hMCM subunits, although at reduced levels (Fig. 2f, lane 1). Thus, the data suggest that hCDT1 and hMCM2-7 do not form a complex in solution but act synergistically during pre-RC formation to promote hMCM2-7 recruitment.

### Human MCM2-7 loading is dependent on hORC1-5, hCDC6 and hCDT1

To address the specificity of hMCM2-7 loading in more detail, we carried out low-salt and high-salt washed pre-RC reactions in the presence of all factors, or with individual factors omitted (Fig. 3a, b). No complex formation was observed without hORC1-5 in low-salt and high-salt conditions, consistent with the concept that hORC1-5 initiates DNA licensing (Fig. 3a, b, lane 1). In the absence of hCDC6, predominantly hORC1-5 is recruited under low-salt conditions (Fig. 3a, lane 2), while hMCM2-7 recruitment was blocked after low and high-salt washes (Fig. 3a, b, compare lanes 2 and 4). Without hCDT1, we observe intermediate hMCM2-7 recruitment and robust hCDC6 recruitment under low-salt conditions (Fig. 3a, compare lanes 3 and 4), while high salt-stable complex formation was blocked (Fig. 3b, lane 3). The high levels of hCDC6 retention suggests that hCDT1 is necessary for hCDC6 release, which can be observed in the presence of the complete reaction.

To obtain insights into the DNA specificity, we asked whether yeast-origin DNA could substitute for human-origin DNA. Therefore, we performed comparative low- and high-salt wash reactions using magnetic beads coupled to either human B2-lamin dsDNA origin sequences or yeast ARS dsDNA origin sequences. However, no differences in hMCM2-7 loading efficiency were observed (Fig. 3c). The data show that high salt stable hMCM2-7 loading is dependent on hORC1-5, hCDC6 and hCDT1 but is independent of human origin sequences.

### hORC6 supports hMCM2-7 double-hexamer formation

Previous work has shown that hORC6 does not form a stable complex with hORC1-5 in the presence or absence of DNA[29,30]. Thus, it was unclear whether hORC6 participates in human DNA licensing at all, and if so, at which step of the process. Though we did not observe hORC6 binding in hORC6-recruitment reactions with hORC1-5 and hORC1-5 and hCDC6 (Fig. 1c), we did observe hORC6 recruitment to hORC1-5, hCDC6, hCDT1 and hMCM2-7 (Fig. 1f) in the presence of ATP after a low-salt wash, thus demonstrating that hORC6 can associate in the presence of all DNA licensing factors. To ask whether hORC6 contributes directly to DNA licensing, we performed low- and high-salt washed reactions in the presence and absence of hORC6 (Fig. 4a, b). The removal of hORC6 did not block pre-RC formation but led to a reduction after the low-salt wash (Fig. 4a, lane 2). Similarly, the high-salt washed complexes showed a 1.8-fold reduction in hMCM2-7 signal (Fig. 4b, lane 2 and Fig. 4c). Thus, we conclude that hORC6 plays a role in supporting complex assembly and high salt-stable hMCM2-7 loading. To further investigate at which step of the helicase loading reaction hORC6 associates, we performed complex assembly reactions with ATPγS. This slowly hydrolysable ATP analogue is known to arrest complex assembly at the OCCM stage in budding yeast. Similarly, here, we could observe the formation of a hOCCM-like complex (Fig. 4d, lane 2). Interestingly, the hORC6 signal was not observed in the presence of ATPγS (Fig. 4d, lane 2), but only with ATP (Fig. 4d, lane 1), suggesting that hORC6 predominantly functions during loading of the second hMCM2-7 hexamer, which in yeast, is recruited following ATP-hydrolysis[63]. In contrast, our data suggest that hCDC6 binding occurs

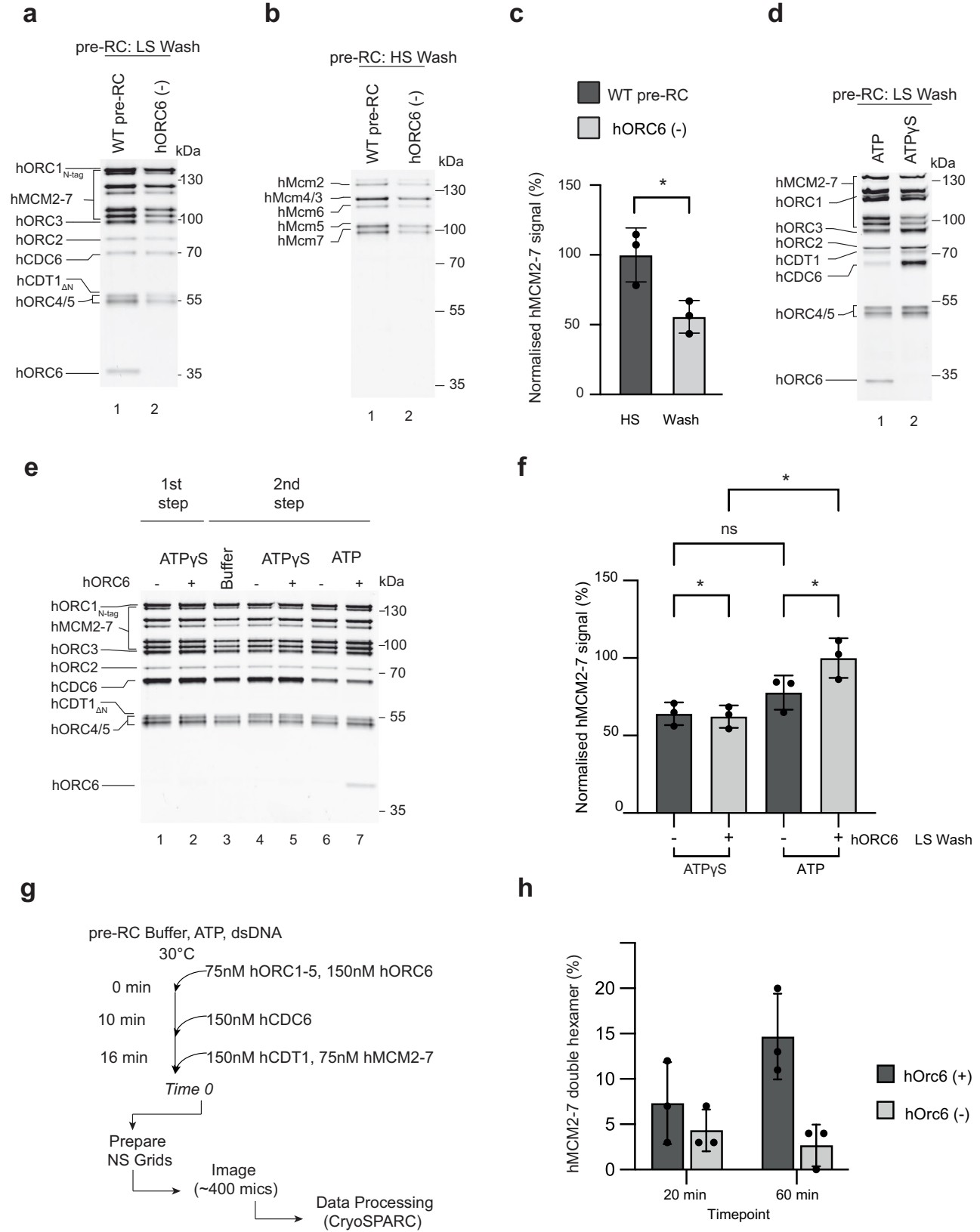

before ATP-hydrolysis, while its release occurs following ATP-hydrolysis. The reciprocal relationship of hCDC6 and hORC6 is consistent with a major reorganisation of the hOCCM following ATP-hydrolysis but may not reflect a direct functional interaction.

Next, we performed staged DNA licensing reactions to further understand the impact of ATP-hydrolysis on hORC6 recruitment and

complex formation. Initially, we assembled reactions with and without hORC6 in the presence of ATPγS for 10 min (Fig. 4e, lanes 1 and 2−first step). We observed identical hOCCM complex formation in both conditions, consistent with the concept that hORC6 recruitment occurs after ATP-hydrolysis. Subsequently, we wanted to better understand the role of hORC6 in the steps after the recruitment of the

**Fig. 4 | The role of hORC6 in DNA licensing.** Specificity of the pre-RC assay was assessed in the presence (WT pre-RC) and absence of hORC6 (hORC6(-)) under (**a**) low and (**b**) high salt conditions. **c** Normalised hMCM2-7 signal in the presence and absence of hORC6 after high salt wash conditions (as in **b**). Mean plotted; individual data points are marked with black circles. $n = 3$ independent experiments, error bars represent standard deviation, statistical significance was calculated using paired, two-tailed $t$ test, *$P \leq 0.05$ ($P = 0.0246$). **d** Pre-RC assays were carried out under low salt conditions in the presence of ATP or ATPγS. **e** Two-step pre-RC assay to assess the timing of hORC6 function carried out in low salt conditions with ATP or ATPγS. The absence and presence of hORC6 is denoted by (−) and (+) respectively. Lane 1 represents the input for the second part of the reactions, shown in lanes 3–7. All SDS-PAGE gels are representative of three independent biological replicates. **f** Normalised hMCM2-7 signal following the two-step pre-RC assay under low salt conditions. Lane 4–7 from (**e**) are quantified. The absence and presence of hORC6 is denoted by (−) and (+) respectively. Mean plotted; individual data points are marked with black circles. $n = 3$ independent experiments, error bars represent standard deviation, statistical significance was calculated using RM one-way ANOVA with Tukey's multiple comparisons test, ns—not significant, *$P \leq 0.05$. $P$ values as presented on graph from left to right: $P = 0.0381$, $0.0995$, $0.0414$, $0.0380$. **g** An in-solution assay was developed to probe double hexamer formation in the presence and absence of hORC6 by negative stain EM. **h** The resulting analysis of double hexamer formation was determined based on 2D class averages resulting after 2 rounds of 2D classification from samples applied onto grids at the indicated timepoints. Mean plotted, individual data points are marked with black circles. Three independent experiments, error bars represent the standard deviation. Source data are provided as a Source Data file.

first hexamer. Therefore, we formed the hOCCM in ATPγS without hORC6 before washing away unbound proteins. To support the recruitment of the second hMCM2-7 hexamer, we added back hCDC6, hCDT1 and hMCM2-7 in reaction buffer with or without hORC6 then allowed the reaction to progress for a further 10 min in ATP or ATPγS. In this second step, we omitted hORC1-5 to suppress new loading events. As a control, we added back ATPγS containing buffer without the additional proteins and observed good hOCCM complex stability (Fig. 4e, lane 3). Under ATP-hydrolysis competent conditions, hORC6 associated with the rest of the pre-RC components, which coincided with additional hMCM2-7 recruitment (Fig. 4e, lanes 6 and 7, Fig. 4f). In contrast, the hORC6-dependent increase in hMCM2-7 signal was not observed in ATPγS (Fig. 4e, lanes 4 and 5, Fig. 4f). Thus, the data suggest that hORC6 functions in the loading of the second hexamer. In principle, this should result in increased hMCM2-7 double-hexamer formation. Since we were concerned about the sliding of hMCM2-7 double-hexamers after the high salt wash, we opted for in-solution DNA licensing reactions which omits the wash step. We performed hMCM2-7 double-hexamer formation in the presence or absence of hORC6 and analysed the reaction products by negative stain-EM after 20-min and 60-min (Fig. 4g). Increased hMCM2-7 double-hexamer formation was observed when hORC6 was included, (Fig. 4h and Supplementary Fig. 3). Thus, the data hint at two different pathways for hMCM2-7 double-hexamer formation: a less efficient hORC6-independent pathway and a more efficient hORC6-dependent pathway. We suggest that the reactions could depend on external factors and biochemical conditions, which will be explored in the future.

## Cryo-EM analysis of the human OCCM
Many of the proteins involved in human helicase loading belong to the AAA+ ATPase family, specifically hORC1, hORC4, hCDC6 and hMCM2-7. Thus, the slowly hydrolysable ATP analogue, ATPγS, can be used to enrich for short-lived early helicase loading intermediates (Fig. 4d). We performed low-salt pre-RC assays with ATPγS and analysed the complexes obtained by negative stain-EM. The resulting 2D class averages were consistent with known reaction intermediates, including hMCM2-7 single hexamers, hOCCM and fully-loaded double hexamers. Nearly half of the particles could be attributed to the hOCCM (Fig. 5a), characterised by a smaller ring corresponding to hORC-hCDC6 which sits atop a larger hMCM2-7 hexamer. As such, the data demonstrate that complex formation was enriched at the hOCCM step and that ATP hydrolysis is important for release of helicase loaders from the helicase.

Next, we aimed to structurally resolve the hOCCM by cryo-EM. The data was collected on a Titan Krios 300 kV microscope and processed using the CryoSPARC2 software suite[64]. The representative 2D class averages are consistent with a hOCCM complex (Fig. 5b, Supplementary Fig. 4b). The final 3D reconstruction from 8730 particles resulted in a structure with an average overall resolution of 6.09 Å (Fig. 5c, Supplementary Figs. 4c and 5, Supplementary Table 1). This allowed for building of a 3D molecular model with good confidence in

backbone trajectories (Fig. 5d). As shown in the local resolution analysis, the upper tier consisting of hORC1-5-hCDC6 is adopting a more fixed conformation, while the lower tier consisting of hCDT1-hMCM2-7 is more flexible (Supplementary Fig. 4e). As expected, the comparison of the hOCCM and yOCCM indicates that the overall organisation is similar (Fig. 5e, Supplementary Fig. 6a, b). Upon closer inspection, we can identify a number of important differences, which start to explain how human DNA licensing differs from yeast DNA licensing.

## hMCM2-7 encircles DNA and adopts an open ring structure
Within the pre-RC intermediate, hMCM2-7 adopts a partially closed ring conformation. The DNA is observed inside the central hMCM2-7 channel; thus, the helicase has already been loaded on DNA (Fig. 5b–d). As in the yOCCM[16], the DNA is not visible within the N-terminal hMCM2-7 ring section (Fig. 5b and Supplementary Fig. 4e, right), likely reflecting an endonucleolytic cleavage event as DNaseI mediated digestion is used to elute protein complexes specifically bound to DNA from the magnetic beads in the pre-RC assay. Consistently, we observe that 38 bp of dsDNA running through the core of the hOCCM has a length similar to the 39 bp observed in the yOCCM structure[16]. The hMCM2-7 ring is partially opened at the hMCM2-hMCM5 gate, and the hMCM5 density is not well resolved, consistent with high local flexibility (Fig. 5c, Supplementary Figs. 4e and 6a). Large sections of the structured N-terminal and C-terminal domains of hCDT1 (aa182-317 and aa413-546) can be seen in the structure, with the N-terminal section of hCDT1 binding to the hMCM2/hMCM6 interface and the C-terminal section to hMCM6/hMCM4, its position within the hOCCM seemingly similar to yCdt1 (Fig. 5d, Supplementary Figs. 4g and 6b). The two domains of hCDT1 are connected by a flexible loop region that was not resolved. Although CDT1 diverged significantly during evolution, with only 10.7% identity and 17.1% similarity between yeast and human CDT1 amino acid sequences, the overall 3D structure and positioning of hCDT1 is conserved from yeast to human. Considering that hCDT1 and hMCM2-7 do not interact in solution, we suggest that initial recruitment of hMCM2-7 to hORC1-5-hCDC6 leads to a reorganisation of hMCM2-7, which then allows hCDT1 to interact with hMCM2-7. What structural change could be involved in the hMCM2-7 reorganisation? We note that the hMCM2-7 N-terminal domains are rotated relative to the C-terminal domain when comparing hMCM2-7[35] and hOCCM, which could support a hCDT1 interaction surface.

## The organisation of hORC1-5-hCDC6 within the hOCCM
The hORC1-5 subunits and hCDC6 are characterised by an AAA+/AAA+-like domain followed by a WHD, with hORC1, hORC2 and hCDC6 carrying N-terminal extensions (Fig. 6a). In the hOCCM complex, hORC1-5-hCDC6 adopt a ring-shaped organisation with DNA located in the centre (Fig. 6b), the complex is organised by an upper C-terminal tier made up from the AAA+/AAA+-like domains and an N-terminal, lower tier made up of the WHDs (Fig. 6b, left). In order to compare the organisation of hORC1-5 in the hOCCM to the hORC1-5-DNA structure (PDB: 7JPS), we aligned both structures via hORC1 (Fig. 6c). This

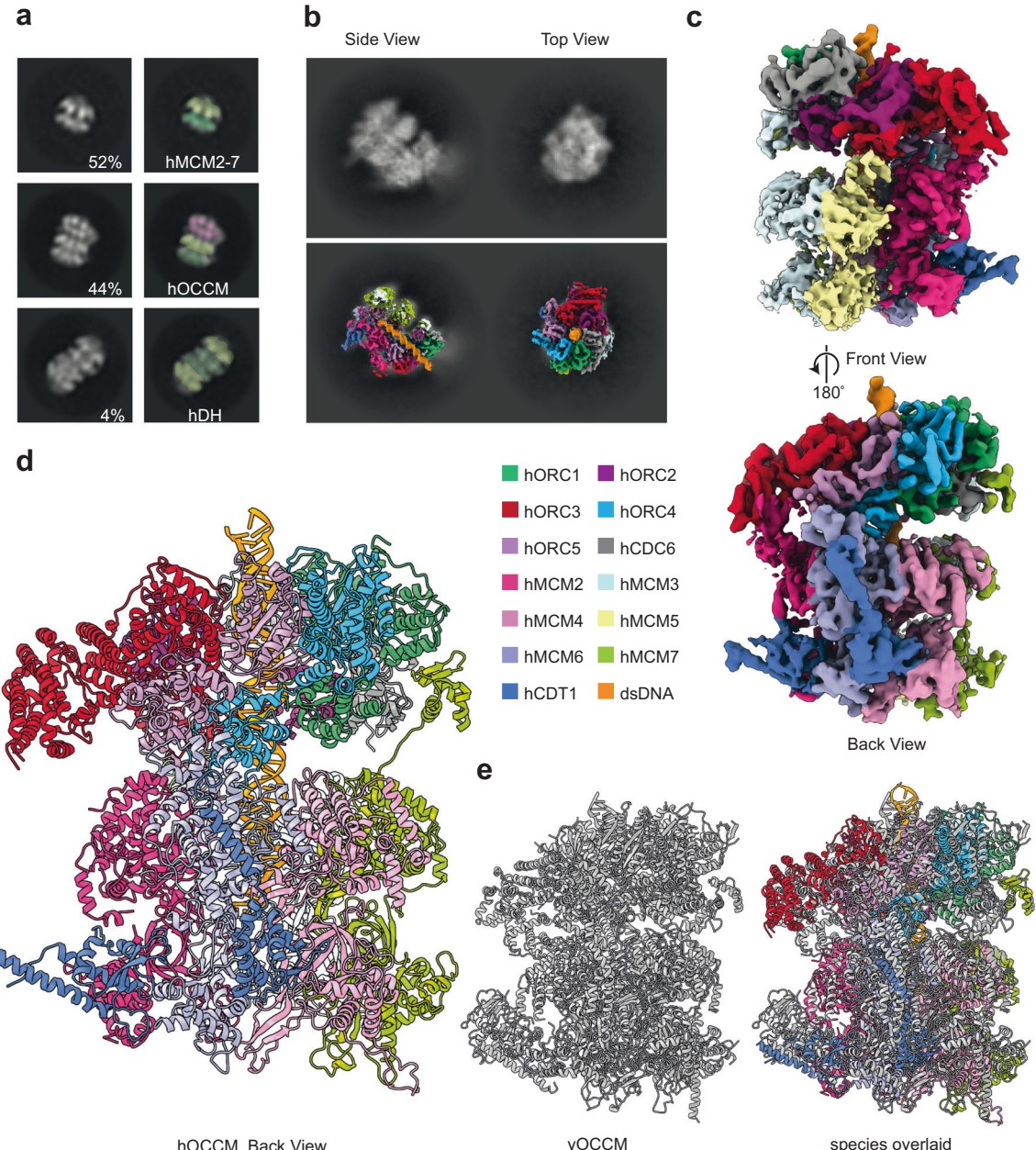

**Fig. 5 | The electron microscopy structure of the hOCCM. a** Negative stain class averages of a low-salt washed pre-RC reaction containing ATPγS. We observed the following pre-RC complexes: single hMCM2-7 hexamers, hOCCM and hMCM2-7 double hexamers. The green densities represent hMCM2-7 rings, shaded for orientation with light green highlighting the C-terminus and darker green N-terminus. hORC-hCDC6 rings have been shaded with pink. **b** Representative 2D class averages from cryo-EM of the hOCCM denoting top and side views (upper), and projections of the resulting 3D map onto the 2D class averages showing good fit (lower). **c** The segmented experimental density map is shown from different angles and coloured by protein subunit. **d** The hOCCM molecular model was built from the cryo-EM structure (PDB: 8RWV). **e** Structure comparison of hOCCM (in colours) with yOCCM (grey, PDB: 5V8F).

highlights that the motor module, consisting of hORC1-hORC4-hORC5, adopts a similar overall conformation. The largest changes in the motor module are observed within hORC5-WHD, which is repositioned in the hORC1-5-DNA structure, likely reflecting that only a very short 13 bp DNA could be observed in the published hORC1-5-DNA complex (Fig. 6d)[31]. Interestingly, more substantial changes can be observed outside of the hORC1-hORC4-hORC5 motor module. Specifically, hORC2 and hORC3 are rotated by 12.8°, which generates a gap between hORC1 and hORC2, that accommodates hCDC6 (Fig. 6e).

We compared available structures from yeast and *Drosophila* in order to localise where hORC6 would dock to hORC3, based on predicted interaction surface[16,65]. We note that while the predicted hORC6 binding surface in hORC3 is accessible, no density for hORC6 was

observed (Fig. 6f). As such, our structural data are consistent with our biochemical observations which showed a lack of hORC6 binding at the hOCCM stage (Fig. 4d, lane 2), which is markedly different from the yeast and *Drosophila* counterparts[16,65]. We speculate that human ORC6 could be recruited in an alternative fashion to the pre-RC, likely involving an interaction with the loaded hMCM2-7 hexamer. Indeed, the N-terminal domain of hMCM6 has been predicted to interact with hORC6[66], very similarly as the yMcm5 interaction with yOrc6[20].

## Role of ATP-hydrolysis in human pre-RC formation

Inspection of the interface of hORC1 and hCDC6 revealed that a loop in hORC1, which has been unstructured in the context of the hORC1-5-DNA structure[31], became structured within the hOCCM and formed a

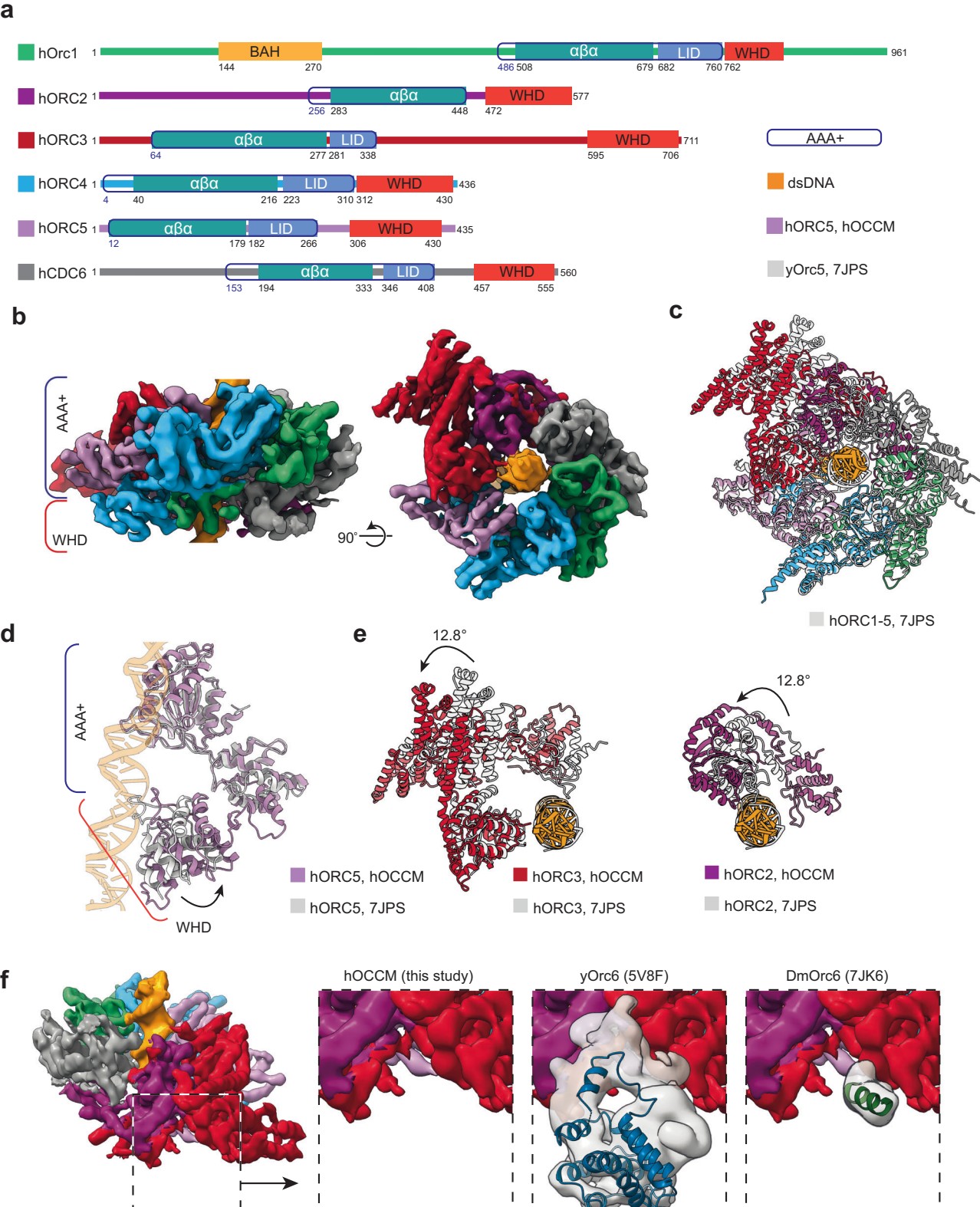

**Fig. 6 | hORC-hCDC6 in the hOCCM structure. a** Domain organisation of hORC1-5 and hCDC6 proteins, and **b** Organisation of the AAA+ and Winged Helix Domains of the DNA-bound hORC1-5-hCDC6 ring in the hOCCM. **c** Comparison of hORC1-5-hCDC6 ring in the hOCCM (coloured) with human hORC1-5-DNA structure (grey, PDB: 7JPS). **d** Reorganisation of hORC5 WHD in hOCCM when compared to hORC1-5-DNA structure (grey, PDB: 7JPS). **e** hORC2 (purple) and hORC3 (red) become reorganised to accommodate hCDC6 in the hOCCM structure compared to the hORC1-5-DNA only structure (grey, PDB: 7JPS). **f** No hORC6 densities were observed in the hOCCM, while yOrc6 was observed in yOCCM (blue, PDB: 5V8F) and DmOrc6 in the active DmOrc1-5 structure (green, PDB:7JK6). Human, yeast and *Drosophila* models were aligned by ORC3. Experimental densities were downloaded from the EMDB and aligned. The Orc6 densities from yeast and *Drosophila* were segmented by a radius of 4 Å and gaussian filtered by 1.5 Å (grey) to display experimental Orc6 position relative to the hOCCM density (coloured).

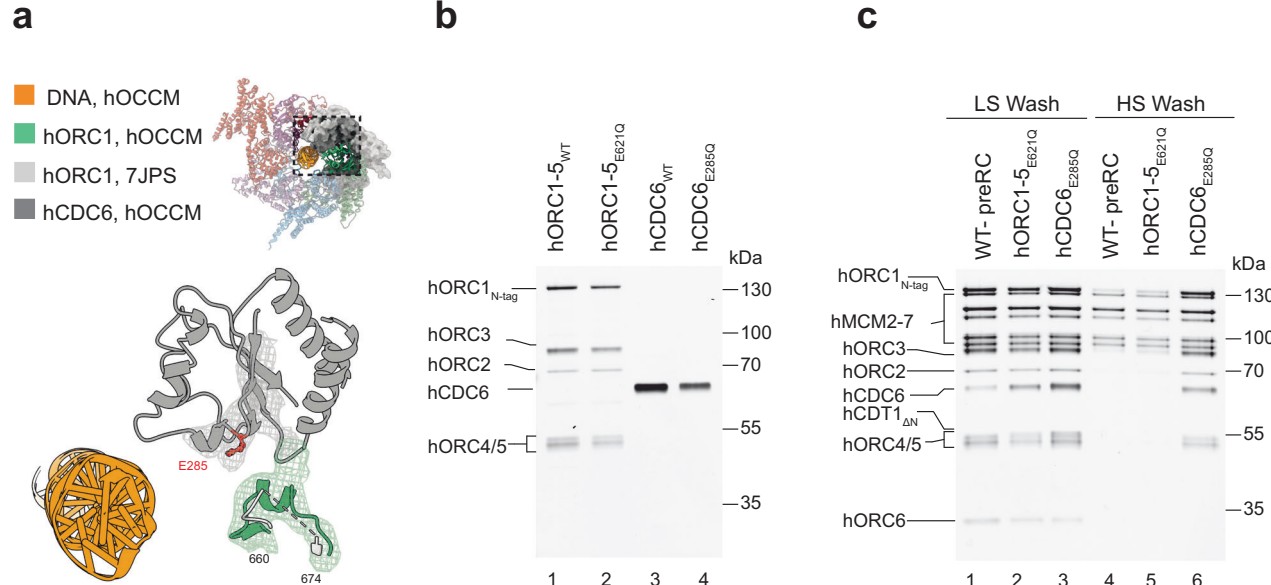

**Fig. 7 | The hORC1-hCDC6 interface becomes restructured in the hOCCM and hCDC6 ATPase is crucial for complex disassembly. a** Observed stabilisation of an hORC1 helix formed between residues 660–674, which is unstructured before hCDC6 (grey) binding as evidenced by docking hORC1 (PDB: 7JPS) in light grey and hORC1 of hOCCM in green. **b** Purified hORC1-5$_{WT}$ and hORC1-5 ORC1 Walker B

mutant (hORC1-5$_{E621Q}$), hCDC6$_{WT}$ and hCDC6 Walker B mutant (hCDC6$_{E285Q}$). **c** The pre-RC assay was carried out under low and high salt wash conditions with hORC1-5$_{WT}$, hORC1-5$_{E621Q}$, hCDC6$_{WT}$ or hCDC6$_{E285Q}$. SDS-PAGE gel is representative of three independent biological replicates. Source data are provided as a Source Data file.

short helix (Fig. 7a). This hORC1 helix contains the conserved arginine finger, which we observed to be proximal to the Walker B motif of hCDC6. It is well established that these two motifs form in AAA+ proteins a composite ATPase site[67]. Thus, the hOCCM structure suggests that hORC1 and hCDC6 form upon complex formation an active ATPase pair. Moreover, hORC1 and hORC4 also form a composite ATPase pair, with a conserved arginine finger in hORC4 and a Walker B motif in hORC1. However, when comparing hORC1-5-DNA with the hOCCM, we did not observe any structural change for the hORC1-hORC4 ATPase pair.

We wanted to investigate whether the observed hORC1 or hCDC6 ATPases function in human pre-RC formation. To that end, we produced Walker B ATPase motif mutants of hORC1 (hORC1-5$_{E621Q}$) and hCDC6 (CDC6$_{E285Q}$) (Fig. 7b, c). Both mutants were tested in the pre-RC assay: after the low salt wash we observed with hORC1-5$_{E621Q}$ reduced complex stability, while hCDC6$_{E285Q}$ led to a stabilisation of hCDC6 (Fig. 7c, lanes 1–3). The even recruitment of hORC6 across the wild-type (WT) proteins and the mutants indicates that complex formation proceeded past the OCCM stage, even in the presence of the ATP hydrolysis deficient hORC1-5 and hCDC6. We next observed that the hORC1 Walker B mutant supported loading of high-salt stable hMCM2-7 complexes onto DNA (Fig. 7c, lane 5). Thus, hORC1 ATPase activity seemingly does not play a major role in DNA licensing. Interestingly, in the presence of hCDC6$_{E285Q}$, hORC1-5-hCDC6 were also stabilised on DNA under high salt wash conditions, highlighting that in fact hCDC6 ATP-hydrolysis is important for hORC1-5-hCDC6 disassembly (Fig. 7c, lane 6). Crucially, in budding yeast none of the yORC, yCdc6 or yMCM2-7 ATPase mutants have led to a stabilisation under high-salt conditions[68]. However, an in vivo analysis of a yCdc6 ATPase mutant revealed that the mutant supported yMCM2-7 binding to origins, but cells remained in S-phase[69]. Interesting, microinjection of an hCDC6 ATPase mutant in human cells in G1 also inhibited DNA replication[70]. As such, these results reveal that hCDC6 ATP-hydrolysis plays an essential role in human pre-RC disassembly, a finding that is consistent with previous cellular experiments in yeast and human.

**hORC-hCDC6-DNA interactions in the hOCCM complex**

The hOCCM structure provides important insights into hORC1-5-hCDC6-DNA interactions, as the DNA extends through the entire hORC1-5-hCDC6 complex. 38 bp of the DNA in the cryo-EM densities were modelled (Supplementary Fig. 4f). When comparing the yeast and human OCCM, it is clear that in yeast, the DNA is bent towards yOrc2 ISM by 10°, while the human DNA is comparatively straight (Fig. 8a). As in the yOCCM, the DNA in the hOCCM was deformed from the standard B-form, with a widening of the major groove and a narrower minor groove (Fig. 8a). While the resolution was limiting for the DNA in the hORC1-5-DNA structure[31], we observed a clear density for the DNA in the hOCCM experimental map (Supplementary Fig. 4f). These results indicate that while hORC1-5 is not known to bind to DNA in a sequence-specific manner, it recognises DNA that is either naturally deformed, or the complex itself introduces structural changes in the DNA upon binding, similar to what has been observed for yORC[37].

Beyond these changes in the DNA structure, in the hOCCM we also observed a series of contacts between hORC proteins and the DNA backbone. Specifically, the hORC and hCDC6 subunits interact with the DNA via two different modules: (1) the initiation-specific motifs (ISMs) in the AAA+ domains and (2) the hairpin and inserted helix in the WHDs (Fig. 8b). Reflecting the 'J' shaped tertiary structure of the individual proteins, ISM contacts localise to the upper part of DNA, toward the N terminus of hORC1-5/hCDC6, while the WHDs exclusively contact DNA more buried in the complex, towards the C-terminus of hORC1-5/hCDC6, which forms interactions with hMCM2-7. Interestingly, the aforementioned structural deformations to the DNA (Fig. 8a) dominantly localise to the more superficial region of DNA that is forming contacts with the ISMs of hORC1-5/hCDC6. As such, the ISMs are responsible for either reading out this structural change, naturally occurring in the DNA, or directly inducing these changes in the structure of the DNA. Upon comparison with the hORC1-5-DNA structure (PDB: 7JPS)[31], we observed additional protein-DNA interactions in the hOCCM. This improved resolution likely reflects an increase in the stability of the DNA in hOCCM, where the length of the DNA is significantly longer.

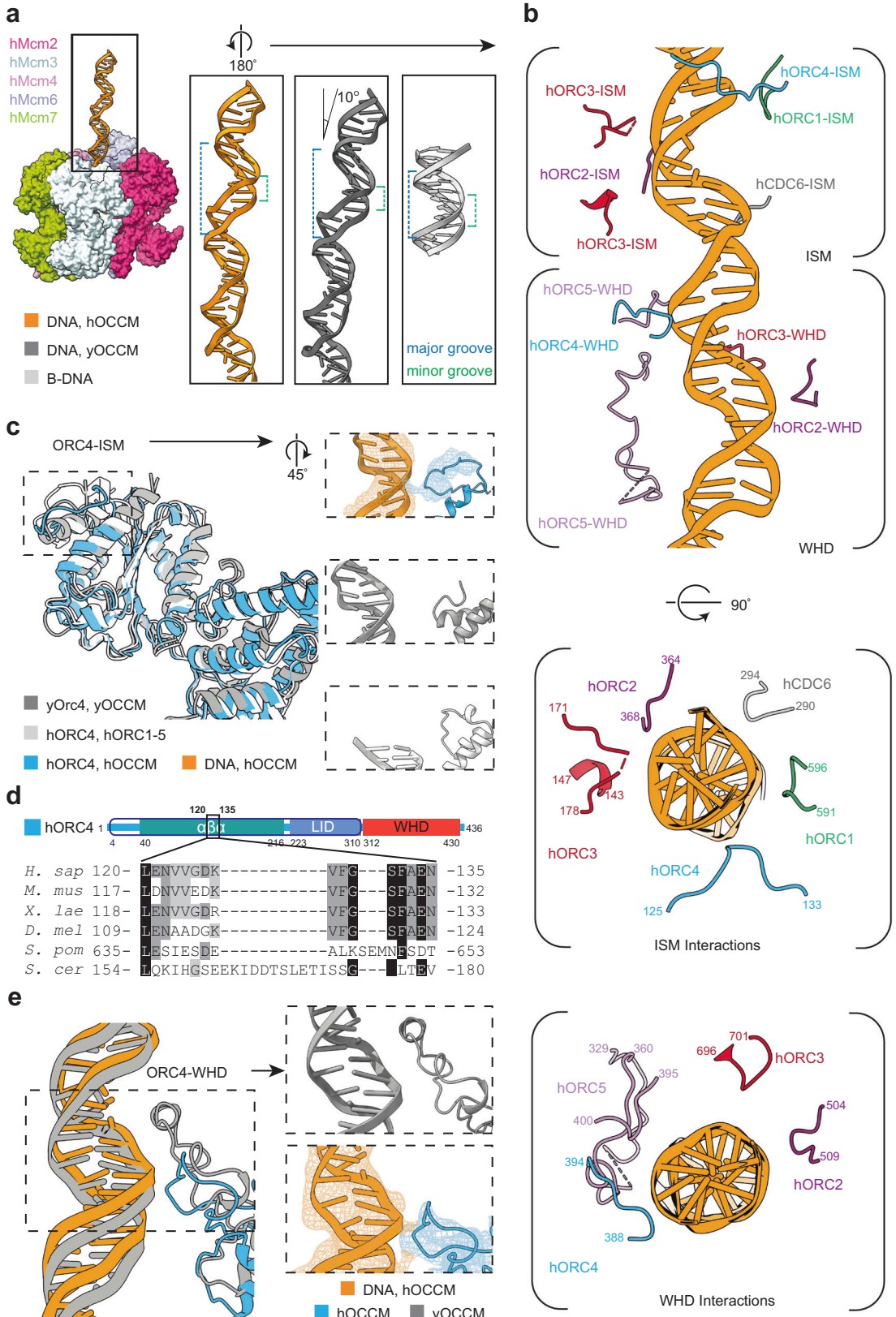

**Fig. 8 | DNA structure and interactions in hOCCM. a** The DNA interacting with yORC in the yOCCM (PDB: 5V8F; grey) is bent by 10° respect to that of the hOCCM (orange). The hOCCM DNA is stretched with respect to the canonical B-form DNA (light grey) in the upper part of the DNA. **b** Detailed hORC1-5 and hCDC6 interactions with DNA. The upper deformed part of the DNA is contacted by the ISM, and the lower less deformed part by WHDs. The view is rotated 180° with respect to (**a**).

**c** hORC4-DNA interactions. hORC4 (light blue) is positioned closer to the DNA in hOCCM than in yOCCM (PDB: 5S8VF; grey). The DNA is not well-resolved enough in hORC1-5 (PDB: 7JPS; light grey). **d** hORC4 sequence alignment. The ISM loop region is longer in yeast containing a helix insert, while higher eukaryotes contain a conserved alternative ISM sequence. **e** Alternative Orc4 WHD close interactions with DNA in the human (blue) and yeast (grey) proteins.

We observed that the hORC1-ISM tracked the phosphate backbone via a loop formed by residues 591–596, in agreement with observations reported from the hORC1-5-DNA structure (PDB: 7JPS) (Fig. 8b and Supplementary Fig. 7a)[31]. By contrast, we observe that the hORC2-ISM, located within the loop spanning residues 364–368 is shifted into the direction of hORC3 and positioned in proximity to the phosphate-backbone of the DNA (Fig. 6e, Supplementary Fig. 7b). This shift represents a difference from the position adopted by the hORC2-ISM loop in the hORC1-5-DNA structure. Moreover, in hOCCM the hORC2-WHD, which was not visible in the hORC1-5-DNA structure, is stabilised. A loop formed by hORC2-WHD residues 504–509 hovers in proximity to the DNA (Fig. 8b, Supplementary Fig. 7c).

Despite discontinuous density in the flexible loop regions of hORC3, a clear interaction is observed between the phosphate-backbones of the DNA and a positively charged ISM loop spanning residues 171–178 (Fig. 8b, Supplementary Fig. 7d). A downstream helix formed by hORC3 residues 143–147, also of the ISM, is slightly more distal to DNA, as are the hORC3-WHD loop residues 696–701 (Fig. 8b, Supplementary Fig. 7e, f). In general, hORC3 contacts appear closer in the context of the hORC1-5 structure than the hOCCM.

hORC4 makes two contacts with the DNA backbone. The hORC4-ISM contains a loop (residues 125–133) that directly binds DNA, differing from the yOCCM and hORC1-5-DNA structures where the loop was oriented more distally to the DNA (Fig. 8c, Supplementary Fig. 7g). A sequence alignment shows that the ISM loop region diverged in higher eukaryotes, reducing a longer loop found in budding yeast (Fig. 8d). Moreover, the hORC4-WHD makes a very close contact with the DNA. Importantly, the hORC4-WHD does not contain the insertion helix which in yOrc4 confers specificity to DNA sequence motifs. Instead, a short loop formed by residues 388–394 of hORC4 interacts with the DNA minor groove, rather than the major groove as for the yOrc4 insertion helix. This human ORC4-WHD loop is well positioned to make phosphate backbone contacts with the DNA, which in yeast is too distal to engage (Fig. 8e, Supplementary Fig. 7h). These observed DNA contacts appear closer in the context of the hOCCM than the hORC1-5-DNA complex, which may reflect that the protein is able to form more stable contacts than could be observed using shorter DNA fragments.

The hORC5-ISM is not in contact with DNA, but the hORC5-WHD makes three contacts with the DNA (Figs. 6d, 8b, Supplementary Fig. 8a). A short loop of hORC5, spanning aa395-400 and aa353-358, is in proximity of both phosphate backbones of a minor groove (Supplementary Fig. 7i, j). The latter forms an extended loop spanning residues 329–360, where hORC5 aa335-339 makes a separate, third contact with the phosphate backbone (Supplementary Fig. 7k). Again, more extended hORC5 DNA contacts have been observed within the hOCCM than in the hORC1-5-DNA complex.

The shorter, partially resolved hORC5 loop spanning residues 395–400 is conserved in both pairwise sequence alignments and molecular localisation with yOrc5 (aa 436-450) (Supplementary Fig. 8b, c). Considering that this loop forms stabilising interactions with both the AAA+ and WHDs of hORC2 in the autoinhibited hORC2-5 complex[26] (Supplementary Fig. 8c), and that the corresponding loop in yOrc5 is crucial in the context of DNA bending[37], this patch may have a regulatory role in DNA licensing.

Notably, we observe residues 335–339 of the long hORC5 loop engaging a region of the DNA nearing the N-terminal of hMCM2-7, which likely only become structured as phosphate backbone contacts are established in the hOCCM structure. Intriguingly, a basic patch formed by this region, conserved in higher eukaryotes, is not observed in sequence alignments or molecular localisation in yeast (Supplementary Fig. 8d and e). Unfortunately, our attempts to replace charged amino acids with alanines in this region resulted in destabilisation of the protein complex.

Finally, the hCDC6-ISM makes contacts the backbone of DNA via a loop formed by residues 290–294 (Fig. 8b, Supplementary Fig. 7l). The hCDC6-WHD is too distant to contact the DNA, owing to a shorter wing in contrast to the yCdc6 WHD. Considering that hCDC6 stabilises hORC1-5 on DNA (Fig. 1c), it is possible that these contacts and the hCDC6 induced reorganisation of hORC2 and hORC3 result in the improved DNA binding. Still, the in vivo relevance of this improved DNA interaction remains to be established.

## Five hMCM WHDs stabilise the hORC1-5-hMCM2-7 interface

At the hORC1-5-hMCM2-7 interface, we observed density for five hMCM C-terminal WHDs, contrasting to only four WHDs which have been reported engaging with yORC in the yOCCM (Fig. 9a).

In agreement with studies carried out in budding yeast, the WHDs of hMCM3 and hMCM7 associate via long loops with hORC2 and hORC1 respectively[16,18]. Unexpectedly, no experimental density was observed in the position occupied by the neighbouring WHD of hMCM4, which was observed in the yOCCM (PDB: 5V8F) (Fig. 9a)[16]. Interestingly, in higher eukaryotes the MCM4-WHD amino acids have diverged (Supplementary Fig. 9), hinting that this WHD may have obtained a different role.

We observed clear densities for the WHDs of hMCM2 and hMCM5, which are involved in interactions with hORC subunits. Yeast does not have a Mcm2-WHD, while the yMcm5-WHD was previously not observed in the context of the yOCCM (Fig. 9a).

To assess the role of the hMCM2 WHD in promoting the licensing reaction, we produced the recombinant helicase incorporating an hMCM2-WHD truncation, missing aa826-904 (Fig. 9c) and tested loading efficiency in the pre-RC reaction. Under low-salt wash conditions, the signal intensity for hCDC6 was slightly enhanced in reactions employing the hMCM2-WHD mutant when compared to reactions carried out using wild type helicase, indicating reduced hCDC6 release (Fig. 9b, compare lanes 1 and 2). After the high salt wash, we observed with the truncation mutant a reduction in high salt-stable hMCM2-7 loading (Fig. 9b, compare lanes 3 and 4). Thus, we conclude that the hMCM2-WHD enhances licensing efficiency, consistent with the concept that multiple hORC-MCM2-7 interactions function synergistically during helicase loading. Importantly, when comparing the hORC1-5-DNA structure with the hOCCM, it is clear that the hCDC6-induced 12.8° rotation of hORC3 enables the hORC3-hMCM2 contact, as this pushes hORC3 further out and generates proximity to the hMCM2-WHD (Fig. 6c, e). Since yeast lacks the entire Mcm2-WHD (Fig. 9c), the evolution of this domain in higher eukaryotes corresponds with enhanced functionality in the licensing reaction, in agreement with our biochemical data. It is also possible that the hMCM2-WHD has additional roles in DNA replication, perhaps serving as a platform for unknown regulators of human DNA replication.

We next attempted to generate a recombinant helicase bearing a hMCM5-WHD truncation spanning 664–734 to test in the pre-RC assay. Unfortunately, purification of this mutant hMCM2-7 complex resulted in poor hMCM5 retention (Fig. 9d), indicating that the hMCM5 WHD contributes to overall complex stability, rendering their analysis in pre-RC formation impossible.

The yMcm6-WHD has a critical role in yOCCM formation, where it functions to regulate DNA insertion and ATP-hydrolysis[7,17,71]. Consistent with this, we observed that the hMCM6-WHD is similarly organised as the yMcm6 WHD. It is sandwiched between hORC4, hORC5 and hCDT1 (Fig. 5c, d and Supplementary Fig. 6b). Specifically, budding yeast Orc4 contains a long loop that is inserted into the interface of the yMcm6 WHD and its C-terminal AAA+ domain (Supplementary Fig. 6b, c). However, no functional relevance for this has been observed in yeast[71]. Consistently, this region is absent in hORC4 (Supplementary Fig. 6d). Instead, the critical MCM6-CDT1 interaction interface is well conserved between budding yeast and human (Fig. 9e). Here, we observed a close contact between hCDT1 (aa413-

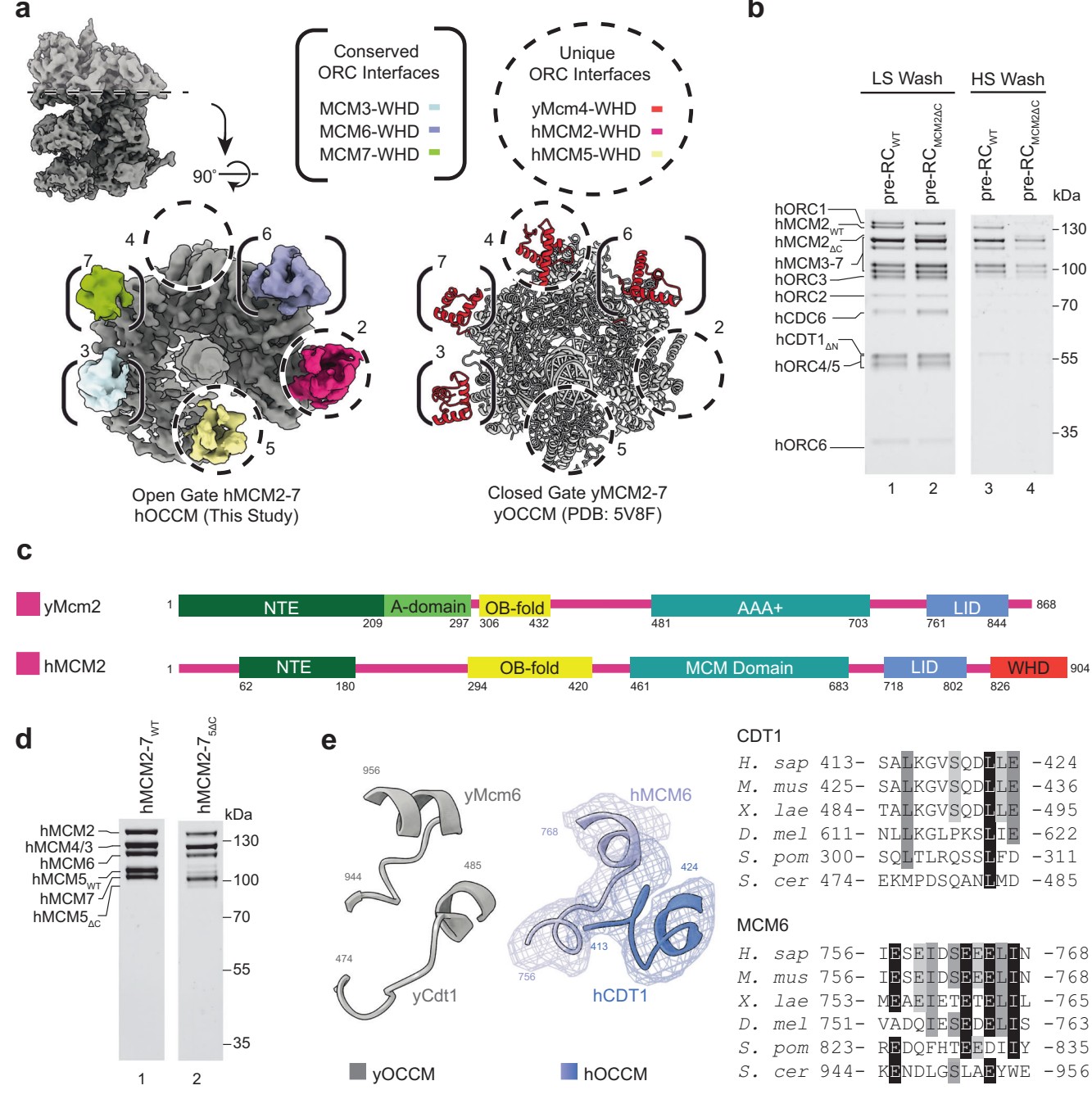

**Fig. 9 | Five hMCM2-7 WHDs make contact with hORC1-5-hCDC6. a** Five hMCM (grey) WHDs (coloured) form contacts with hORC1-5-hCDC6 in hOCCM, contrasting with only four inter-subunit interactions in yOCCM (PDB: 5V8F). The conserved interfaces are highlighted in brackets, and the unique interfaces are circled. **b** Pre-RC assay under low and high salt wash conditions was carried out with WT and hMCM2$_{\Delta C}$ (hMCM2-WHD truncated mutant) hMCM2-7. SDS-PAGE gel is representative of two independent biological replicates. **c** Comparison between yeast and human MCM2 domain organisation. hMCM2$_{\Delta C}$ lacks aa826-904, the WHD domain (red). **d** Purification of the hMCM5$_{\Delta C}$ truncation mutant results in sub-stoichiometric hMCM2-7 complexes compared to purified hMCM2-7$_{WT}$. **e** Structural comparison of the human and yeast MCM6-CDT1 interface and sequence alignment of the corresponding region. Source data are provided as a Source Data file.

424) and hMCM6 (aa756-768). Previously, hMCM6 E757 and L766 and hCDT1 R425, I426, K429 and K433 were identified to be important in pull down assays[62,72]. However, in this NMR analysis (PDB: 2LE8), the short hCDT1 peptide has substantially shifted[72] when compared to the hOCCM structure, indicating that structural context is necessary to fully understand the interaction. Thus, the hOCCM structure can now explain how hCDT1 and hMCM6 interact in the context of FL proteins within the hOCCM complex.

## Structural changes in hORC2 connect hORC1-5 auto-inhibition to hMCM3-WHD recruitment

Crucially, the AAA+ domain of hCDC6 stabilises the hORC2-WHD in the hOCCM, thus revealing its active conformation (Fig. 10a, left). In contrast, in hORC2-5[31] the hORC2-WHD was folded inwards blocking DNA interactions (Fig. 10a, centre), something also seen in the *Drosophila* Orc1-6 complex. (Fig. 10a, right)[65]. This rotation of the ORC2-WHD defines the autoinhibited state, wherein the ORC complex cannot bind

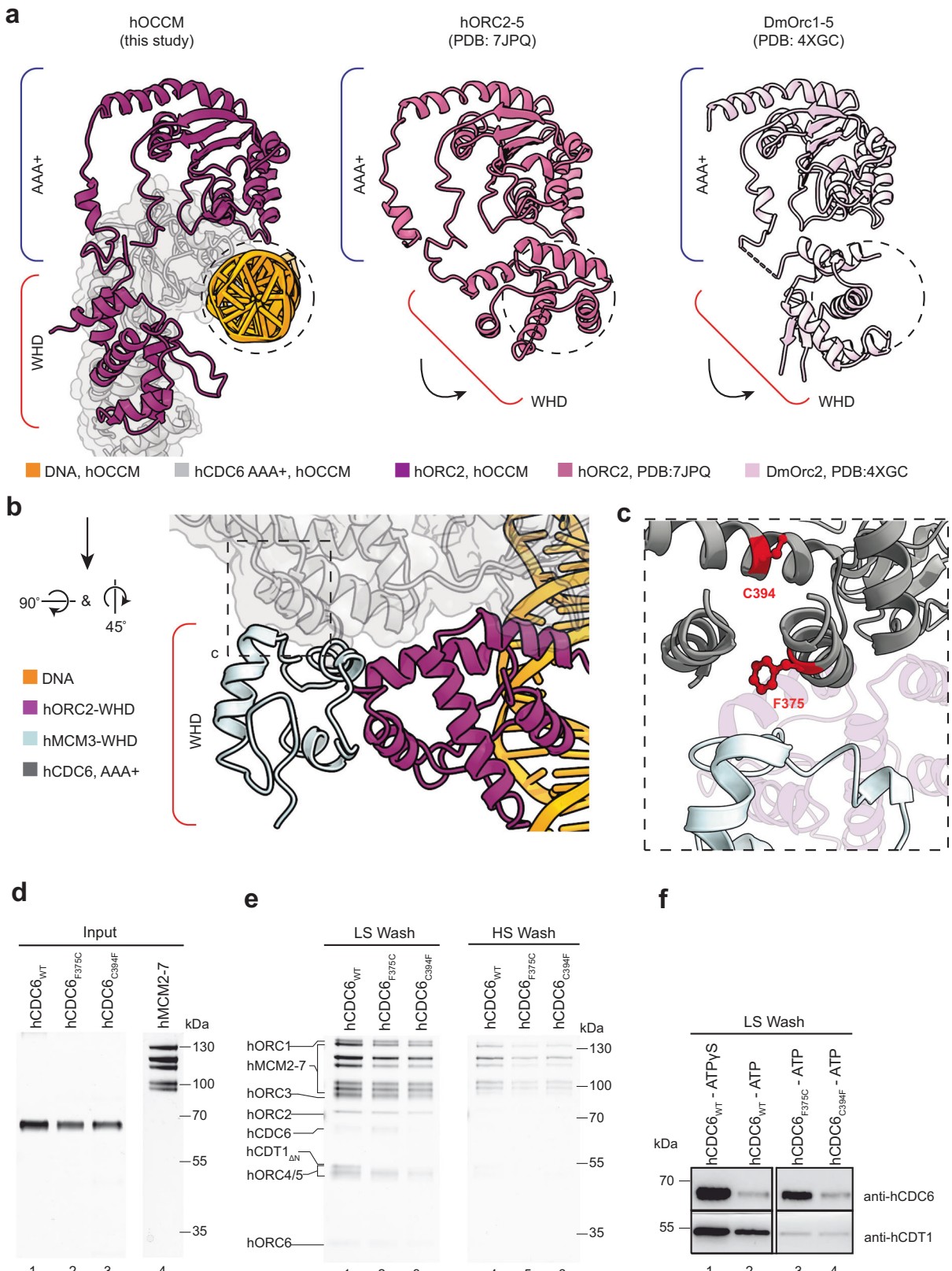

DNA. Consistently, without hCDC6 stabilising the hORC2-WHD, the flexible hORC2-WHD was not resolved in the hORC1-5-DNA structure[31]. Crucially, the hORC2-WHD conformation observed here is specifically recognised by hMCM3 (Fig. 10b). This data brings to light the critical role hCDC6 plays in positioning the hORC2-WHD, to activate the

hORC1-5 complex and facilitate the observed hORC2-WHD-hMCM3 contacts.

In budding yeast, it is well established that the yMcm3-yCdc6 interaction initiates pre-RC formation. Our data show that the hCDC6 AAA+ lid and WHD participate in the interaction with hMCM3

**Fig. 10 | The hORC2-WHD in the hOCCM is reorganised upon DNA and hCDC6 binding. a** Structure of hORC2-WHD. The hORC2 (mauve) is stabilised in the open claw position by the presence of hCDC6 AAA+ domain (light grey), as compared with the closed claw conformations of hORC2-WHD in hORC2-5 (PDB: 7JPQ) and in DmOrc2 (PDB: 4XGC) which is not compatible to forming an interaction with DNA. **b** The orientation adopted by the hORC2-WHD (mauve) upon hCDC6 binding falls within the vicinity of the hMCM3-WHD (light blue) in the hOCCM structure. **c** The location of relevant COSMIC[55] mutations (red) in hCDC6 (grey) in the hOCCM structure may locally disrupt interaction with hMCM3 via disruptions to inter- and intra-protein interactions. **d** Purified hCDC6$_{WT}$, hCDC6$_{F375C}$ and hCDC6$_{C394F}$. **e** Pre-RC assay under low and high salt wash conditions was carried out with hCDC6$_{WT}$, hCDC6$_{F375C}$ and hCDC6$_{C394F}$. SDS-PAGE gel is representative of four independent biological replicates. **f** Corresponding western blot analysis for hCDC6 (anti-hCDC6 mouse monoclonal, Santa Cruz, Cat# sc-9964) and hCDT1 (anti-hCDT1 mouse monoclonal, Santa Cruz, Cat# sc-365305) in the presence of hCDC6$_{WT}$, hCDC6$_{F375C}$ and hCDC6$_{C394F}$. ATPγS arrests complex formation at the hOCCM stage and shows maximal hCDC6 and hCDT1 association. Western blots are representative of two independent biological replicates. Source data are provided as a Source Data file.

(Figs. 5c, d and 10c). Interestingly, inspection of the Catalogue Of Somatic Mutations In Cancer (COSMIC) revealed 83 missense mutations in this region of hCDC6. Structural analysis of these mutations using MutaBind2[73] highlighted that only few of these mutations are predicted to have a detrimental phenotype. F375C has been found in a lymphoid neoplasm and C394F in adenocarcinoma and non-small cell carcinoma. While hCDC6 F375 is localised directly at the interface with the hMCM3 WHD, hCDC6 C394 is pointing into the inner core of the hCDC6 AAA+ lid domain (Fig. 10c). We purified these hCDC6 mutants (Fig. 10d) and asked whether they impact DNA licensing.

Both mutants led to a reproducible reduction in DNA licensing, visible under low- and high- salt wash conditions (Fig. 10e, lanes 2–3 and 5–6). Specifically, C394F led to diminished hCDT1 and hMCM2-7 recruitment, as observed after a low salt wash and reduced high salt stable hMCM2-7 loading, but did not impact on hCDC6 recruitment (Fig. 10f and Supplementary Fig. 10). Due to the introduction of a bulky hydrophobic amino acid into the core of the hMCM3-WHD, it is likely that this mutation is affecting the stability of hMCM3-WHD domain, hence resulting in random unfolding events that reduce activity, but do not block one specific step in complex formation. With F375C, we observed reduced hCDT1 and hMCM2-7 recruitment after the low-salt wash and reduced high salt-stable hMCM2-7 loading (Fig. 10f and Supplementary Fig. 10). In contrast, hCDC6 was stabilised (Fig. 10f). Since hCDC6 release is ATP-hydrolysis dependent (Fig. 10f, compare lanes 1 and 2), the data suggest that this mutant affects a step prior to ATP hydrolysis, consistent with a specific defect in hMCM3 recruitment. In summary, our data demonstrate that cancer-associated mutations can impair DNA licensing in vitro and that the assay can provide initial insights into their mechanism of action. Although hMCM2-7 loading was impacted by the hCDC6 mutation, the cancer cells were growing[55]. This could be because most cancer-associated mutations are heterozygous and, thus, mutations can be tolerated. However, reduced hMCM2-7 loading would render these cells more prone to genome stability and sensitise them to chemotherapy.

In summary, we have developed a powerful reconstituted system that enables the mechanistic, structural and functional analysis of DNA licensing (Fig. 1). Related work was developed by colleagues at the same time[74,75]. Our assay is highly specific (Fig. 1f) and results in the assembly of salt-stable hMCM2-7 double-hexamers (Figs. 1–3), the final product of the DNA licensing reaction. We show that yeast and human MCM2-7 DNA licensing results in complexes of similar salt stability (Fig. 2a). Using the assay, we identify that hORC6 does not participate in the loading of the first hMCM2-7 hexamer, but in the loading of the second double-hexamer (Fig. 4e–h). This is markedly different from budding yeast, where yOrc6 forms an integral part of the complex and is involved in DNA bending during yORC-yCdc6-DNA complex formation[13]. This bending step is critical during DNA licensing, as it generates space for yMCM2-7 recruitment[17]. Future structural work has the potential to reveal if and how hORC6 independent DNA bending occurs during hORC1-5-hCDC6-DNA complex formation. Interestingly, while yCdt1 forms a stable complex with yMCM2-7, hCDT1 and hMCM2-7 do not interact in solution (Fig. 2d, e). Consistently, hCDT1 is not essential for initial hMCM2-7 recruitment (Fig. 2f). However, hCDT1 plays an essential role for ATP-hydrolysis dependent hCDC6 release (Figs. 3a and 4d). The structure of the hOCCM provided deep insights

into the overall complex organisation (Fig. 5). We showed that hCDC6 alters the organisation of hORC1-5, which supports in turn the establishment of multiple interactions with hMCM-WHDs (Fig. 6). Moreover, our structure-function analysis uncovered how the hMCM2-WHD, which does not exist in yeast, acts in human DNA licensing (Fig. 9). Our structure provided a snapshot of the active hORC1-hCDC6 ATPase interface. Interestingly, by analysing a hCDC6 ATPase mutant we discovered that in the absence of hCDC6 ATPase a rare high salt stable helicase loading intermediate is formed (Fig. 7c), highlighting that hCDC6 ATPase activity is essential for to resolve this highly stable complex. In contrast, yCdc6 ATPase activity has only a role in quality control, e.g. when the complex becomes phosphorylated[7,18]. Finally, we used the assay to uncover the impact of cancer-associated mutations in hCDC6 and revealed that selected mutants at the hCDC6-hMCM3 interface led to a reduction in DNA licensing activity (Fig. 10). Whether or not this finding is of functional impact to patients is currently unknown but will be of interest in the future. The established licensing system will be key to generate future mechanistic insight into the human DNA replication, will provide insights into disease causing mutations and is well suited to understand the regulation of the DNA licensing process.

## Methods

### Expression and purification of wild type and mutant hORC1-5

hORC1-5 expression plasmids were generated by gene synthesis (Genscript) based on pESC vectors (Stratagene) for the galactose inducible overexpression in budding yeast; yielding pCS1026, containing ORFs for WT hORC5 and hORC1 as an N-terminal fusion protein with a Strep tag, pCS1366 containing ORFs for WT hORC5 and hORC1 E621Q Walker B mutant as an N-terminal fusion protein with a Strep tag, pCS1099 containing the WT hORC2 ORF, and pCS1103, encoding the FL WT hORC3 and hORC4 ORFs. The proteins were expressed in *S. cerevisiae* YC658 cells (MATa, lys2::pGAL1 GAL4::LYS2, pep4::HIS3, bar1::hisG derived from W303)[76] using established conditions[24]. The cells were broken using a SPEX™ freezer-mill. Lysates were resuspended in lysis buffer (50 mM HEPES-NaOH (pH 7.6), 250 mM KCl, 50 mM NaCl, 2 mM DTT, 5% (w/v) Glycerol, 2 mM MgCl$_2$, 0.2% (w/v) NP-40, 10 mM B-glyceroPO$_4$) supplemented with benzonase (Merck), complete EDTA-free protease inhibitor tablets (Roche) and PhosSTOP phosphatase inhibitor cocktail tablets (Roche). Clarified lysates were then loaded onto a StrepXT column (Cytiva) followed by a wash with lysis buffer, and elution using lysis buffer supplemented with 75 mM biotin. Subsequently, eluted protein was supplemented with 2 mM MgCl$_2$, 1 mM ATP, and diluted to 200 mM NaCl for ion exchange chromatography. In some cases, eluates were incubated with SUMOstar protease (LifeSensors) overnight to remove N-terminal Strep tag on hORC1. Subsequently, protein was bound to a 1 mL HiTrap Heparin column (Cytiva). The column was washed with low salt buffer (50 mM HEPES-NaOH (pH 7.6), 200 mM NaCl, 1.5 mM DTT, 25 mM NaF, 5% (w/v) Glycerol, 2 mM MgCl$_2$, 0.02% (w/v) NP-40), before being eluted using a salt gradient increasing the buffer to 1 M NaCl. Peak fractions were pooled and concentrated for loading onto a 10–30% 400 μL glycerol density gradient prepared in 20 mM HEPES-NaOH (pH 7.6), 300 mM NaCl, 1 mM DTT, 2 mM MgCl$_2$, 1 mM ATP and glycerol. Gradients were subjected to ultracentrifugation at 102,000 × *g* for 16 h. After manual gradient fractionation, SDS-PAGE and Coomassie staining was used to

identify fractions containing stochiometric protein. Finally, target fractions were pooled, concentrated and buffer-exchanged into a final buffer of 20 mM HEPES-NaOH (pH 7.6), 300 mM NaCl, 1 mM DTT, 2 mM MgCl$_2$, 1 mM ATP and <10% glycerol, using a 0.5 mL Amicon® Ultra Centrifugal Filter (50 kDa MWCO). Concentrated protein aliquots of ~3–9 μM were snap-frozen in LN$_2$ and stored at −80 °C.

### Expression and purification of hORC6

The hORC6 expression plasmids were generated by gene synthesis (Genscript) based on pET21a for the IPTG inducible overexpression in bacteria. Plasmid pCS1049, coding for FL, WT hORC6 with an N-terminal His tag, was transfected into *Escherichia coli* (*E. coli*) Rosetta 1 (DE3) cells (Agilent) and grown in LB media supplemented with ampicillin. Cells were inoculated at an OD$_{600}$ of 0.05 and grown at 37 °C to mid-log phase, at which point protein expression was induced by the addition of IPTG. Cells were harvested after 16 h of protein expression at 16 °C by centrifugation at 3500 × *g* for 10 min. Cell pellets were resuspended in lysis buffer (50 mM HEPES (pH 7.5), 150 mM NaCl, 10 mM imidazole, 5% (v/v) glycerol, 1 mM DTT) and lysed by sonication on ice (Branson Digital Sonifier 450/120C). The clarified lysates were then loaded onto a HisTrap Excel column (Cytiva) equilibrated with lysis buffer. The column was washed first using 5 column volumes (CV) of lysis buffer, then 15 CV of a high salt buffer (lysis buffer supplemented with 1 M NaCl) followed by additional 30 CV lysis buffer. Protein was eluted from the column using lysis buffer supplemented with 450 mM imidazole and incubated overnight with PreScission protease (Cytiva) at 4 °C to cleave the His-tag. Subsequently, eluted protein was bound to a POROS HQ column equilibrated in buffer A (30 mM HEPES-NaOH (pH 7.5), 150 mM NaCl, 1 mM DTT) for anion exchange chromatography. Protein was eluted across a gradient increasing Buffer B (30 mM HEPES-NaOH (pH 7.5), 1 M NaCl, 1 mM DTT) to 70% over 15 min. Peak fractions were collected and loaded onto a HiLoad Superdex 75 16/60 column equilibrated in buffer C (10 mM HEPES-NaOH (pH 7.5), 150 mM NaCl, 1 mM DTT) for size exclusion chromatography. Purified hORC6 protein was pooled, concentrated in a 30 kDa MWCO Amicon® Ultra-4 concentrator (Millipore), aliquoted and flash-frozen in LN$_2$.

### Expression and purification of wild type and mutant hCDC6

The hCDC6 expression plasmids were generated by cloning the human *CDC6* gene into pGEX6P1 (Cytivia) for the IPTG inducible overexpression in bacteria. Plasmids for WT hCDC6 (pCS536), hCDC6 Walker B mutant E285Q (pCS1339), hCDC6 cosmic mutant F375C (pCS1418) and hCDC6 cosmic mutant C394F (pCS1419) were transfected into *E. coli* BL21 codon + RIL cells (Agilent) and grown in LB media supplemented with ampicillin and chloramphenicol. Cells were inoculated at an OD$_{600}$ of 0.05 and grown at 37 °C to mid-log phase, at which point protein expression was induced by the addition of IPTG. Cells were harvested after overnight protein expression at 16 °C by centrifugation at 3500 × *g* for 10 min. Cell pellets were resuspended in lysis buffer (50 mM HEPES-NaOH (pH 7.6), 250 mM KCl, 50 mM NaCl, 2 mM MgCl$_2$, 0.02% (v/v) NP40, 10% (v/v) Glycerol, 2 mM DTT) and lysed by sonication (Branson Digital Sonifier 450/120C). Clarified lysates were incubated with GST-Agarose resin (Sigma) for 1 h at 4 °C on a rotator. The resin was washed with 10 CV lysis buffer supplemented with 1 mM ATP, 10 CV High Salt buffer (50 mM HEPES-NaOH (pH 7.6), 1 M KCl, 50 mM NaCl, 2 mM MgCl$_2$, 0.02% (v/v) NP40, 10% (v/v) Glycerol, 1 mM DTT, 1 mM ATP) and 3 CV buffer C (30 mM HEPES-NaOH (pH 7.6), 167 mM KCl, 33.2 mM NaCl, 1 mM MgCl$_2$, 0.02% (v/v) NP40, 5% (v/v) Glycerol, 1 mM DTT, 1 mM ATP). Finally, 2 CV Buffer C was added to the resin along with PreScission protease (Cytivia). After O/N cleavage at 4 °C, the eluted protein was loaded onto a HiTrap SP HP column (Cytiva) equilibrated in buffer C for cation exchange chromatography. Protein was eluted across a gradient increasing

buffer D (30 mM HEPES-NaOH (pH 7.6), 825 mM KCl, 165 mM NaCl, 1 mM DTT, 5% (w/v) Glycerol, 1 mM MgCl$_2$, 0.05% (v/v) NP-40) to 100%. Purified hCDC6 protein was pooled, concentrated in a 30 kDa MWCO Amicon® Ultra-4 concentrator (Millipore), aliquoted and flash-frozen in liquid nitrogen.

The Walker B mutant hCDC6 was further purified by gel filtration using a HiLoad Superdex 200 16/60 column. The COSMIC hCDC6 mutants were concentrated directly following elution from the affinity resin using 30 kDa MWCO Amicon® Ultra-15 Centrifugal Filter and buffer exchanged into final storage buffer (10 mM HEPES- NaOH pH 7.6, 250 mM KCl, 50 mM NaCl, 1 mM DTT) before being aliquoted and flash frozen without further purification.

### Expression and purification of wild type hCDT1

The hCDT1 expression plasmids were generated by gene synthesis (Genscript) and pGEX6P1 (Cytivia) was employed for the IPTG inducible overexpression in bacteria. Plasmid pCS1149, coding for FL hCDT1 fused to an N-terminal GST tag, were transfected into BL21 (Agilent) and grown in LB media supplemented with ampicillin. Cells were inoculated at an OD$_{600}$ of 0.05 and grown at 37 °C to mid-log phase, at which point protein expression was induced by the addition of IPTG. Cells were harvested after overnight protein expression at 16 °C by centrifugation at 3000 × *g* for 20 min. Cell pellets were resuspended in lysis buffer (50 mM HEPES-NaOH (pH 7.6), 250 mM NaCl, 2 mM DTT, 0.1% (v/v) Triton X-100, 10% (v/v) glycerol) and lysed via sonication (Branson Digital Sonifer 450/120C). The clarified lysate was added to pre-equilibrated Sepharose Glutathione Fast-Flow resin (Sigma) and incubated at 4 °C for 2 h. PreScission protease (Cytivia) was added and the solution was incubated overnight at 4 °C. The protein was eluted by washing with buffer B (30 mM HEPES-NaOH (pH 7.6), 200 mM NaCl, 1 mM DTT, 0.01% (v/v) Triton X-100, 5% (v/v) glycerol) then was loaded onto a POROS™ HS 20 μm column (ThermoFisher) and eluted using a 200 to 1000 mM NaCl gradient. The protein was concentrated using a 30 kDa MWCO Amicon Ultra-4 concentrator (Millipore) and further purified by gel filtration chromatography on a Superdex 200 Increase 10/300 GL column (Cytiva) equilibrated in final storage buffer (10 mM HEPES-NaOH (pH 7.6), 200 mM NaCl, 1 mM DTT, 0.01% (v/v) Triton X-100, 5% (v/v) glycerol). Protein peak fractions were pooled, concentrated, aliquoted and flash-frozen in liquid nitrogen.

Plasmid pCS1273, based on pET21A (+), coding for N-terminally truncated hCDT1$_{158-546}$ fused at the N-terminus to a 6xHis tag, was generated by Genscript and was transfected into BL21 (DE3) and grown in LB media supplemented with ampicillin. Cells were inoculated at an OD$_{600}$ of 0.1 and grown at 37 °C to mid-log phase, at which point protein expression was induced by the addition of IPTG. Cells were harvested after overnight protein expression at 16 °C by centrifugation at 3000 × *g* for 50 min. Cell pellets were resuspended in lysis buffer (50 mM HEPES-NaOH (pH 7.6), 200 mM NaCl, 10 mM imidazole, 2 mM DTT, 0.02% (v/v) NP-40, 10% (v/v) glycerol) and lysed via sonication (Branson Digital Sonifer 450/120C). The clarified lysate was added to a pre-equilibrated HisTrap Excel column (Cytivia). The column was washed with 10 CV lysis buffer before the protein was eluted using a 10–450 mM imidazole gradient in lysis buffer. The eluate was dialysed overnight in buffer B (30 mM HEPES-NaOH (pH 7.6), 200 mM NaCl, 1 mM DTT, 0.02% (v/v) NP-40, 5% (v/v) glycerol) then was loaded onto a POROS™ HS 20 μm column (ThermoFisher) and eluted using a 200–1000 mM NaCl gradient in buffer B. The protein was concentrated using a 30 kDa MWCO Amicon Ultra-4 concentrator (Millipore) and further purified by gel filtration chromatography on a Superdex 200 Increase 10/300 GL column (Cytiva) equilibrated in final storage buffer (10 mM HEPES-NaOH (pH 7.6), 200 mM NaCl, 1 mM DTT, 0.02% (v/v) NP-40, 5% (v/v) glycerol). Protein peak fractions were pooled, concentrated, aliquoted and flash-frozen in liquid nitrogen.

## Expression and purification of wild type and mutant hMCM2-7

hMCM2-7 complexes were expressed in HEK293-F cells by Oxford Expression Technologies Ltd (Oxford, UK) as described before[77]. WT hMCM2-7 was expressed from plasmids pCS1247 (containing ORFs for N terminal twin Strep tagged-hMCM2 hMCM3 and hMCM5) and pCS1248 (containing ORFs for N terminal 8xHis tagged-hMCM4, hMCM6 and hMCM7). Mutant hMCM2(1-868) and hMCM5 (1-663) were encoded on plasmids pCS1406 and pCS1408, respectively.

The cell mass was resuspended in 25 mL lysis buffer (20 mM HEPES-NaOH (pH 7.5), 250 mM KGlu, 5 mM MgCl₂, 2 mM ATP, 10 mM imidazole, 0.02% (v/v) NP-40, 5% (v/v) glycerol) containing one SIGMAFAST™ EDTA-free protease inhibitor tablet (Sigma) and Benzonase (Millipore). Cells were lysed via sonication (Branson Digital Sonifer 450/120C). The membrane fraction was removed by centrifugation at $41,000 \times g$ for 50 min loaded onto Ni-NTA agarose (Qiagen, #30210) with pre-equilibrated lysis buffer. The resin was washed with 5 CV lysis buffer with protease inhibitor and 5 CV lysis buffer without protease inhibitor. The protein was then eluted using 2 CV elution buffer (20 mM HEPES pH 7.5, 250 mM KGlu, 5 mM MgCl₂, 2 mM ATP, 300 mM imidazole, 0.02% (v/v) NP-40, 5% (v/v) glycerol) before loading onto a StrepTrapXT column (Cytiva) equilibrated with buffer B (20 mM HEPES-NaOH (pH 7.5), 250 mM KGlu, 5 mM MgCl₂, 2 mM ATP, 1 mM DTT, 0.02% (v/v) NP-40, 5% (v/v) glycerol). The column was washed with 3 CV buffer B before protein was eluted using 3 CV buffer B supplemented with 50 mM biotin. Fractions were visualised with SDS-PAGE, pooled and concentrated using a 50 kDa MWCO Amicon Ultra-15 centrifugal concentrator (Millipore). Purified protein was aliquoted, flash frozen in LN₂ and stored at −80 °C.

## Pulldown assay

Purified hORC1-5 (47 nM), hORC6 (47 nM), hCDC6 (94 nM) were mixed in pulldown buffer (15 mM HEPES-KOH pH 7.5, 200 mM NaCl, 1 mM DTT, 2 mM MgCl₂,1 mM ATP, 0.1% (v/v) Triton) in the presence or absence of 90 bp yeast ARS1 dsDNA containing the A, B1 and B2 elements (Sequence listed in supplementary data 1). After addition of Mag Strep type 3 XT beads (IBA Lifesciences) and a 10 min incubation at 22 °C the beads were washed 3 times. Consequently, the proteins were eluted with biotin (2 mg/ml). The eluted proteins were separated by SDS-PAGE and silver stained.

## Human pre-RC assay

For the human DNA licensing assay, purified hORC1-5 (47 nM), hORC6 (94 nM), hCDC6 (94 nM), hCDT1 (47 nM) and hMCM2-7(70 nM) were incubated in pre-RC buffer (25 mM HEPES-KOH (pH 7.5), 250 mM KGlu, 1 mM DTT, 4 mM MgOAc, 0.1% (v/v) Triton-X100, 1 mM ATP or ATPγS) for 10 min at 30 °C with shaking. A 2kbp or 3kbp fragment of human B2-lamin dsDNA was amplified by PCR from template pCS1087 (primer sequences listed in supplementary data 1). Next, 600 ng of linear human B2-lamin dsDNA, either 2kbp or 3kbp in length, conjugated to magnetic streptavidin beads (MyOne Streptavidin T1. Invitrogen) via 5' biotinylation, was added to the reaction which was incubated for an additional 20 min under the same conditions. The magnetic beads are washed twice with either a low salt buffer (pre-RC buffer) to remove excess unbound protein, or a high salt buffer (pre-RC buffer supplemented with 300 mM NaCl) to disrupt hydrophilic interactions and leave only those complexes bound and directly encircling the DNA. Finally, DNA-bound complexes are eluted from the beads by briefly incubating in an elution buffer prepared from pre-RC buffer supplemented with 5 mM CaCl₂ and the endonuclease DNaseI (Thermo Scientific). Eluates were subjected to either SDS-PAGE or used directly to prepare cryo-EM grids.

To assess hORC1-5 binding to human B2-lamin DNA fragments, 300 bp fragments of human B2-lamin dsDNA were amplified by PCR from templates pCS1192-1201 (primer sequences listed in supplementary data 1). 2600 ng of linear 300 bp human B2-lamin fragment dsDNA conjugated to magnetic streptavidin beads via 5' biotinylation were used in the assay.

To assess human pre-RC assembly on non-human DNA, a 3 kbp fragment of ARS1 DNA was amplified by PCR from template pCS372 (primer sequences listed in supplementary data 1) and conjugated to magnetic beads in the same way as human B2-lamin 2kbp/3kbp DNA.

For the staged pre-RC, reactions were assembled as described in ATPγS containing pre-RC buffer. 10 min after the addition of DNA, reactions were washed twice with low salt buffer (ATPγS) before resuspension in pre-RC buffer containing hCDC6 (94 nM), hCDT1ΔN (47 nM) and hMCM2-7 (70 nM) and 1 mM ATP or ATPγS. hORC6 (94 nM) was either included or omitted from the reaction. After an additional 10 min, beads were washed twice with low salt buffer containing ATP or ATPγS, before proceeding to elution as described.

## In solution pre-RC assay

For the in-solution assembly of human DNA licensing products, purified hORC1-5 (75 nM), hORC6 (150 nM) and a 1.2 × M excess of 158 bp lamin B2 dsDNA (90 nM) (DNA sequence derived from MCM ChEC-Seq analysis[78] is shown in supplementary data 1) were first incubated in pre-RC buffer (25 mM HEPES-NaOH (pH 7.5), 250 mM KGlu, 1 mM DTT, 4 mM MgOAc, 0.05% (v/v) Triton-X100, 1 mM ATP) for 10 min at 30 °C with shaking at 600 RPM. After 10 min, purified hCDC6 (150 nM) was added to the reaction, which was maintained at 30 °C with shaking. At the same time, purified hMCM2-7 was pre incubated at 30 °C prior to addition to the reaction. This step serves to in order to destabilise cold-induced dimerisation of purified hMCM2-7 single hexamers. Finally, purified hMCM2-7 (75 nM) and hCDT1ΔN (150 nM) were added to the reaction, 6 min after the addition of hCDC6. The reaction was left to proceed at 30 °C with shaking for up to 60 min following the addition of all components. Samples were taken at 20 min and 60 min and used directly for negative stain EM.

## Negative stain electron microscopy

**Grid staining.** For in solution pre-RC assays, 3.5 µL of reactions containing 75 nM hORC1-5, 150 nM hORC6, 150 nM hCDC6, 150 nM hCDT1 and 75 nM hMCM2-7 in pre-RC Buffer (25 mM HEPES-NaOH (pH 7.5), 250 mM KGlu, 1 mM DTT, 4 mM MgOAc, 0.05% (v/v) Triton-X100, 1 mM ATP) were applied to carbon film 300 mesh Cu grids which had been made hydrophilic by glow discharging using a PELCO easiGlow device. Samples were incubated for 30 s before blotting, and subsequently washed twice using pre-RC buffer and once using milli-Q grade water. Grids were then stained twice with 2% uranyl acetate.

For all pre-RC assay samples, 12 µL eluates from the DNA licensing assay containing -100 ng of DNA-protein complexes were applied to carbon film 300 mesh Cu grids which had been made hydrophilic by glow discharging using a PELCO easiGlow device. Samples were incubated for 30 s before blotting, and subsequently washed twice using pre-RC buffer and once using milli-Q grade water. Grids were then stained twice with 2% uranyl acetate.

**Data collection.** Data were collected with a Thermo Fisher Talos F200i TEM operated at 200 kV. The microscope was equipped with a Falcon 3EC direct electron detector. EPU software was used for automated acquisition of micrographs collected at x73 k magnification using a defocus range from −0.9 to −2.9 µm. Magnification settings resulted in a pixel size of 2.0 Å.pixel⁻¹.

**Image processing.** In all instances, micrographs were imported into CryoSPARC v4.03. Particles were picked from 10−100 micrographs using CryoSPARC own implementation of blob picker, followed by inspect particle picks and filtered using NCC score and local power. Following particle extraction and iterative 2D classification, 10−20 representative 2D class averages were used for template picking on the complete dataset. For yeast double hexamers (Fig. 2c) 934

micrographs were collected. Initially, 248,212 particles were extracted into 200 × 200 pixel size boxes and subjected to 2D classification. After pruning, the resulting 143,947 particles were subjected to 2D classification and subset selection. Double hexamers accounted for 91% or 130,356 particles. The remaining 12,616 or 9% of particles corresponded to smaller, single hexamer class averages.

For human double hexamers (Fig. 2c) 615 micrographs were collected. Initially, 8120 Particles were extracted into 200 × 200 pixel size boxes and subjected to 2D classification. After pruning, the resulting 7431 particles were subjected to 2D classification and subset selection. Double hexamers accounted for 57% or 4233 particles. The remaining 3189 or 43% of particles corresponded to smaller, single hexamer class averages.

For hOCCM (Fig. 5a), 915 micrographs were collected. Initially, 140,082 particles were extracted into 200 × 200 pixel size boxes and subjected to 2D classification. After pruning, the resulting 81,890 particles were subjected to 2D classification and subset selection. Class averages corresponding to hOCCM, recognisable by a distinct multimeric assembly, accounted for 44% of the data or 36,363 particles. Double hexamers accounted for only 4% or 3079 particles. The remaining 52% or 42,448 particles corresponded to smaller single hexamer assemblies including MCM2-7 top views. The resulting hOCCM 2D class averages were used as a template for picking during subsequent high-resolution cryo-EM data processing.

### Cryo-EM analysis
**Sample vitrification.** For all samples, ~100 ng of DNA licensing intermediates were applied to 2 nm Quantifoil R2/2 holey carbon support grids which had been glow discharged using a PELCO easiGlow. After 30 s, grids were blotted for 1 s before being plunged into liquid ethane kept at liquid nitrogen temperature for vitrification, using a TFS Vitrobot Mark IV under 95% humidity.

**Data collection.** Data were automatically collected with EPU software on a Titan Krios TEM (Thermo Fisher Scientific) operated at 300 keV and equipped with a K3 direct electron detector. The total dose was ~40 electrons per Å$^2$ for a total of 40 frames per micrograph (Supplementary Table 1). A total of 18,728 micrographs was collected with a defocus range between −0.5 to −2.1 μm. Magnification settings resulted in a pixel size of 1.1 Å.pixel$^{-1}$.

**Image pre-processing.** Original image stacks were summed and corrected for drift- and beam-induced motion using CryoSPARC Patch Motion Correction[64]. The estimation of contrast transfer function (CTF) parameters for each micrograph was performed with PatchCTF[64].

**Reconstruction.** Image processing was carried out using CryoSPARC v4.03 or Relion v4.0 (see Supplementary Fig. 5). For CryoSPARC processing the micrographs were split into 4 subsets and processed in parallel for particle picking and 2D classification, before being merged for 3D reconstructions. Class averages of hOCCM particles from cryo-grid screening data, resolved using blob picker, were used as templates for another particle picking round and low-pass filtered to 50 Å. Picks were pruned to exclude areas containing thick carbon and extracted into 300 × 300 pixel size boxes (1.1 Å.pixel$^{-1}$ sampling) giving a total number of 268,243 particles which were included as input for an Ab initio reconstruction. High resolution particles were sorted from low resolution particles in multiple rounds of heterogeneous 3D refinement. Finally, 8730 particles were included in a non-uniform 3D refinement to generate a consensus map with a global resolution reported as 6.09 Å. Half maps, refined maps, sharpened maps and masks used have been deposited to the EMDB (EMD-19566).

**Model building.** AlphaFold models for human DNA licensing proteins were rigid-body docked into the locally refined maps of the hOCCM using UCSF Chimera v1.16[79]. This served as a starting point for molecular model building in COOT v0.9.7[80], which was combined iteratively with refinement using Namdinator[81] and Phenix v10.20.1[82]. The molecular model was subjected to geometry minimisation. The molecular model was deposited to the PDB: 8RWV.

### Mass photometry
Mass Photometry measurements were carried out using a TwoMP Mass Photometer (Refeyn) at room temperature. Samples were added to a six well silicone gasket (GraceBioLabs) on a pre-cleaned sample coverslip (Refeyn). Movies were recorded for 60 s in the regular field of view using the AcquireMP software (v. 2023R2, Refeyn Ltd), and analysed using the DiscoverMP software (v. 2023R2, Refeyn Ltd). Prior the measurements, BSA (Sigma, 66 kDa monomer, 132 kDa dimer) and bovine thyroglobulin (Sigma, 670 kDa) were measured to produce a linear mass calibration. The maximum mass error for each calibration was below 5%. Measurements were completed in Mass Photometry buffer (pre-RC buffer without Triton-X100). For the hMCM2-7/hCDT1 and yMCM2-7/yCdt1 measurements, 75 nM MCM2-7 was incubated with 150 nM CDT1 (750 nM for the 10x hCDT1 measurement) for 20 min at 30 °C in Mass Photometry Buffer. The instrument was focused using the Droplet Dilution routine in the AcquireMP software with the samples diluted 1 in 10 on the instrument. Measurements were repeated in triplicate. Measurements of the other pre-RC proteins were completed as described above but with the proteins diluted to 75 nM and immediately measured.

### Quantification of pre-RC silver stain bands
Multi Gauge V2.3 (FujiFilm) was used to measure the density of each hORC1-5 or hMCM2-7 protein band from silver-stained pre-RC SDS-PAGE gel. All densities were corrected against a background control from the same lane. The mean density of hORC1-5 and hMCM2-7 protein bands for each pre-RC reaction were calculated. Mean and standard deviation values were determined from at least three independent experiments. Data was normalised, plotted and analysed (Two-tailed paired $t$ test or RM one-way ANOVA with Tukey's multiple comparisons test) using GraphPad Prism (Version 10.2.2).

### Reporting summary
Further information on research design is available in the Nature Portfolio Reporting Summary linked to this article.

## Data availability
The map of the hOCCM have been deposited in the EMBD with accession codes EMD-19566, and the atomic models in the Protein Data Bank under accession codes PDB-8RWV. Requests for resources and reagents should be directed to and will be fulfilled by the Lead Contact Christian Speck (chris.speck@imperial.ac.uk). Source data are provided with this paper.

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

## Acknowledgements

We thank members of the Speck lab for critical reading of the manuscript. We thank Audrey Mossler for help with expression and purification of proteins and Anja Schlott for cloning WT hCDC6. We thank Luca Pellegrini for the human MCM expression plasmids. The work was supported by Cancer Research UK (DRCNPG-May21\100006) and the Wellcome Trust (107903/Z/15/Z) with both grants awarded to C.S. We thank the LMS and ICL cryo-EM facilities for support. We thank the LonCEM cryo-EM facility funded by Wellcome Trust (206175/Z/17/Z) and Nora Cronin for support with the data collection.

## Author contributions

J.N.W., L.V.E., V.L. and C.S. conceived the project and designed the experimental approaches. V.L. established the DNA licensing assay, initial protein expression and purification conditions. J.N.W., V.L., L.V.E., S.A., M.P., J.T., A.K.S. and C.M.P. optimised protein purification and purified proteins. J.N.W., L.V.E., V.L., M.P., J.T., S.V.F. and L.R. performed the biochemical assays. S.A. and L.V.E. designed ATPase mutant experiments. J.T. and L.V.E. carried out statistical analysis of biochemical assays. M.P. carried out Mass Photometry experiments. J.N.W. and V.L. and characterised protein complexes for cryo-EM structural determination and performed initial sample screenings. J.N.W. prepared cryo-EM samples and performed cryo-EM imaging with support/input from R.A. J.N.W. processed cryo-EM data with input from A.K.S. and R.A. A.K.S, J.N.W. and R.A. built the atomic models. J.N.W. and C.S. interpreted the structural data. J.N.W. and C.S. prepared figures. J.N.W. and C.S. wrote the manuscript with critical input from all authors. C.S. and V.L. acquired funding for this study.

## Competing interests

The authors declare no competing interests.
