## [Transparent Peer Review file · Nature Communications]

Reconstitution of human DNA licensing and the structural and functional analysis of key intermediates

Corresponding Author: Professor Christian Speck

Version 0:

Reviewer comments:

Reviewer #1

(Remarks to the Author)

Helicase loading, also named pre-replication complex assembly and DNA licensing is a multi-step process that occurs prior to cell division, when the genome needs to be duplicated. In their manuscript, Wells and colleagues aim at reconstituting the human pre-replication complex in vitro by using purified proteins. This complex is analyzed both functionally by pull-down assays performed in low- and high- salt conditions, and structurally by electron microscopy. This work focuses on DNA licensing in human but integrates a detailed comparison with the budding yeast, which is the best described eukaryotic system for the DNA replication machinery. Generally, the manuscript is well-written, and the figures are very nice and informative. This work includes numerous results that will contribute to a better understanding of DNA licensing in humans and the phenotypes found in cancers and other human pathologies.

However, the authors should address the following points:

Major points:

1) The authors report the cryo-EM 3D reconstruction of the human ORC-Cdc6-Cdt1-MCM2-7 complex at 6.1 Å resolution. To make myself clear, I think that important conclusions can be made from this reconstruction reported at such an intermediate resolution. However, the authors should clarify how model building was performed from AF predictions. In its current form the model building method section is insufficiently detailed. In particular, it would be important to show the quality of the map surrounding the regions of the OCCM that are interacting with the DNA (possibly, in a supplementary figure). The authors should clarify what can be seen and what cannot in the model. (e.g. page 10 "large sections of hCdt1 can be seen in the structure, ..." is too vague. Some statements are somehow overstated given the resolution of the structure: Page 12 ("phosphate-backbone contacts with the DNA" & "... hovers in proximity to the DNA without making direct contacts" this cannot be stated confidently at this reported resolution, direct contacts may exist but cannot be resolved? I would recommend being careful about drawing conclusions on the chemistry of the interactions if the side chains cannot be resolved); Page 13 ("hOrc5 tracks both phosphate backbones" hOrc5 could not interact with the bases as well ?)

2) This model was calculated using only the best 8730 particles from a large initial dataset including 18022 movies. The image processing workflow should provide more details regarding the hetero refinement and 3D classifications step. The data were processed using Cryosparc. Given the heterogeneity of the sample, were alternative EM reconstruction programs used, such as Relion for example?

Minor points:

- 1) Page 7, line 4 : "hCdt1 that hOrc1-5" should be "hCdt1 and hOrc1-5"
- 2) Please make sure that Mcm or MCM are homogeneously defined through the manuscript (e.g. page 9 – line 32, Mcm2-7 should be MCM2-7).
- 3) Page 9-line 32: ... likely due to DNase mediated digestion" this sentence will sound confusing for readers, which may not be familiar with the DNase1 treatment used in the pre-RC assay. Please revise.
- 4) The method section does not mention how many times the pull-down experiments shown in figure 1 for example were replicated independently.
- 5) Figure 2, panels c and d: the method section should mention how many movies were collected for the two datasets and how many particles were used to determine the percentages DH vs SH.
- 6) Supplementary figure 7: the figure is nice and informative. However, mapping the conserved residues on a 3D structure

will be useful. The Mcm4 sequences diverge but there are also conserved residues. The authors could consider showing a surface representation using consurf?

7) Figure 8: number is not correct in the alignment shown panel e (956 not 756 for Mcm6 from *S. cerevisiae*)

Reviewer #2

(Remarks to the Author)

In the present manuscript, Jennifer et al. report the *in vitro* reconstitution of human DNA replication licensing using purified components (MCM2-7, ORC1-5, ORC6, CDC6 and CDT1). This assay is used to compare human and yeast DNA licensing system. In contrast to yeast, hOrc6 is found to be inessential for recruitment of the first MCM2-7 hexamer while hCdt1 does not show a tight association with hMCM2-7 in solution and is also found to be inessential for the initial recruitment of hMCM2-7. In addition to reconstitution assays, a structure of hOCCM is also solved, providing insights into the overall complex organization.

Overall, the work is of high significance and interest to readers from multiple fields. The field of eukaryotic DNA replication initiation has long focused on yeast as a model system, but recent evidence has indicated that metazoan DNA replication systems may have important differences. The findings systematically recapitulate DNA licensing process with human factors, considerably advancing the field's understanding of this process. Providing the presentation of data can be better organized (especially the somewhat haphazard callout of figure panels) and a few extant questions concerning the approaches resolved, publication would seem warranted.

Primary comments:

1. Fig. 1c and 1g. It is claimed that there are differences in the binding efficiency between Orc1-5 and DNA +/- Cdc6 and in Mcm2-7 loading efficiency +/- Orc6. The gels suggest this is true, but the differences are too subtle to be sure. Quantitation of the species across replicates would be helpful.
2. On page 7 it's said that "hOrc6 predominantly functions during loading of the second MCM2-7 hexamer", but it's not explicated shown whether hOrc6 is required for this event. Biochemical assays and negative stain EM should be included to answer this question.
3. Perhaps it was missed during reading, but in Fig. 3c, please explain why the amount of hCdc6 decreases in lane 2 compared to lane 1. Is hCdt1 promoting ATP hydrolysis and hCdc6 release?
4. On a related note, why does the binding hOrc6 and hCdc6 to hOrc1-5 seem to be reciprocally antagonistic (e.g., Fig. 1h and 3c)? Is hCdt1 and/or Cdc6 ATPase activity controlling hOrc6 association?
5. In previous study (ref. #27), hORC subunits were found to form a complex in an ATP-dependent manner. It may be worth testing whether hOrc6 binds to hOrc1-5 in the presence of ATP at low salt (not 200mM NaCl), since hOrc6 was seen to remain bound to preRC under these conditions in the presence of ATP (Fig. 1h).
6. Please state the protein concentrations used for the pre-RC formation and pull-down experiments.
7. The binding of hCdt1 to hMcm2-7 is ruled out using mass photometry; however, this was done at very low concentrations of hMcm2-7 (~7.5nM) and hCdt1 (~15nM) and in the absence of ATP. Binding should be examined at a higher protein concentration and in the presence of ATP.
8. Fig. 6d. It is surprising that the ATPase activity of Orc1 is not necessary for Mcm2-7 loading. However, the experiment compares a single WB mutant in Cdc6 to a double WB Orc1 and RF Orc4 mutant. Does the Cdc6 WB mutant behave differently if it is paired with an RF Orc1 mutation?
9. The build for the DNA looks relatively poor, with a fair degree of backbone and base distortions (as evidenced by the buckling of the cartoon rendering). When building and refining into low resolution maps, it is important that restraints (strong geometry weighting, maintaining appropriate H-bonding distances between base pairs and in secondary structural elements) be applied and enforced. The model should be reexamined and refined accordingly.
10. Fig. 8a. The observation that a different cohort of Mcm WHDs contact ORC-Cdc6 is interesting. Could this be because the Orc1-5/Cdc6 ring is cracked (as shown in the figure) for human but not yeast? And why is the ring cracked in one system but not the other? (It would be helpful to keep the rendering style the same between the two proteins.)
11. Given the low resolution of the maps, it is important to show the density associated with all claimed interactions. (e.g., Figs. 6a, Figs. 7e, SF5c, SF6c)

Minor points:

12. Please provide references for the last couple of sentences in the 2nd paragraph of the Introduction.
13. For Fig. 1g, a panel of LS washes should be added as a control.
14. It would be helpful to expand the discussion concerning the role of hCdt1, instead of only one conclusion "Thus, the data suggest that hCdt1 and hMCM2-7 do not form a complex in solution but act synergistically during pre-RC formation to promote hMCM2-7 recruitment." The structural analysis of hCdt1 is also largely missing, only the interface between hMcm6 and hCdt1 is discussed.
15. hCdt1 didn't co-purify with MCM2-7, but hCdt1 is solved in the OCCM complex structure. Does hCdt1 only interact with loaded MCM2-7?
16. In the second paragraph of section "hOrc1-5 binding to DNA is stabilised by hCdc6", it is stated that "While hOrc1-5 only associated with dsDNA alone and only background binding of hOrc6 was observed (compare lane 4 with lanes 2 and 3)" The figure panel showing these lanes should be noted in the text.
17. The callout of Figure 5 in section "hMCM2-7 encircles DNA and adopts an open ring structure" seems to have some inconsistencies. For example, Fig. 5d doesn't seem to show that "DNA is not visible within the N-terminal Mcm2-7 ring section". And from Fig. 5c it cannot be concluded that the "hMcm2-7 ring is partially opened at the Mcm2-Mcm5 gate". Fig. 5d also doesn't show yCdt1 for comparison with hCdt1.
18. The title of Fig. 6 seems wrong? It's not Cdc6 that is restructured but hOrc1?
19. For the statement, "In contrast, in hOrc2-530 the hOrc2-WHD was folded inwards blocking DNA interactions (Fig. 9a,

- centre), something also seen in the Drosophila Orc1-6 complex", a citation is needed for the final clause.
20. Fig. 5a. Please add a color key for identifying subunits.
21. In figure 7c, hOrc2 loop starts and ends with the same residue number (364).
22. On page 15, "Unfortunately, purification of this mutant MCM2-7 complex resulted in poor Mcm5 retention (Fig. 8)". Here the Fig. 8 should be Fig. 8d.
23. On page 15, it is stated "Consistent with this, we observed that the hMcm6-WHD is similarly organised as the yMcm6 WHD. It is sandwiched between Orc4, Orc5 and Cdt1 (Fig. 5c, d, Supplementary Fig 5b)." Fig. 5c,d doesn't show hMcm6-WHD.
24. On page 15, it is stated "Here, we observed a close contact between hCdt1 (aa413-424) and hMcm6 (aa756-786)". Should '756-786' be '756-768'?
25. At the end of page 16, "Consistent with the essential role of Mcm3 in DNA replication and cellular viability, the impact of the mutations was not severe." Why is Mcm3 essential, but the mutations don't have severe consequences?
26. Fig. SF1d. Why are there multiple masses for MCM2-7?
27. In Fig. S4, the iterative Homo/Hetero Refinement and Subset Selection step is confusing. Please simplify this.
28. It would be appropriate to mention the other two preprints posted online recently that are pertinent to the work:
(<https://www.biorxiv.org/content/10.1101/2024.04.10.588796v1>,
<https://www.biorxiv.org/content/10.1101/2024.04.10.588848v1>)

Version 1:

Reviewer comments:

Reviewer #1

(Remarks to the Author)

The authors have adequately addressed all of my concerns. Actually I commend them for being thorough and thoughtful in their replies. Well done.

Reviewer #2

(Remarks to the Author)

All issues have been addressed. Thank you!

We appreciate the reviewers' comments and suggestions and are grateful for their insightful input, which has helped us enhance the quality and clarity of our work.

Reviewer #1 (Remarks to the Author):

Helicase loading, also named pre-replication complex assembly and DNA licensing is a multi-step process that occurs prior to cell division, when the genome needs to be duplicated. In their manuscript, Wells and colleagues aim at reconstituting the human pre-replication complex in vitro by using purified proteins. This complex is analyzed both functionally by pull-down assays performed in low- and high- salt conditions, and structurally by electron microscopy. This work focuses on DNA licensing in human but integrates a detailed comparison with the budding yeast, which is the best described eukaryotic system for the DNA replication machinery. Generally, the manuscript is well-written, and the figures are very nice and informative. This work includes numerous results that will contribute to a better understanding of DNA licensing in humans and the phenotypes found in cancers and other human pathologies.

We want to thank the reviewer for the kind words.

However, the authors should address the following points:

Major points:

1) The authors report the cryo-EM 3D reconstruction of the human ORC-Cdc6-Cdt1-MCM2-7 complex at 6.1 Å resolution. To make myself clear, I think that important conclusions can be made from this reconstruction reported at such an intermediate resolution. However, the authors should clarify how model building was performed from AF predictions. In its current form the model building method section is insufficiently detailed. In particular, it would be important to show the quality of the map surrounding the regions of the OCCM that are interacting with the DNA (possibly, in a supplementary figure). The authors should clarify what can be seen and what cannot in the model. (e.g. page 10 "large sections of hCdt1 can be seen in the structure, ..." is too vague. Some statements are somehow overstated given the resolution of the structure: Page 12 ("phosphate-backbone contacts with the DNA" & "... hovers in proximity to the DNA without making direct contacts" this cannot be stated confidently at this reported resolution, direct contacts may exist but cannot be resolved? I would recommend being careful about drawing conclusions on the chemistry of the interactions if the side chains cannot be resolved); Page 13 ("hOrc5 tracks both phosphate backbones" hOrc5 could not interact with the bases as well?)

We appreciate the reviewer's suggestion. Accordingly, we have described the model building in more detail (see Material and Methods) and generated several additional map-to-model comparisons (e.g., Supplementary Fig. 7). Also, we have clarified what can be seen in the structure and revised the statements about interactions, as suggested.

2) This model was calculated using only the best 8730 particles from a large initial dataset including 18022 movies. The image processing workflow should provide more details regarding the hetero refinement and 3D classifications step. The data were processed using Cryosparc. Given the heterogeneity of the sample, were alternative EM reconstruction programs used, such as Relion for example?

Thank you for the suggestions. We have provided a new image processing workflow incorporating more details on the requested steps (Supplementary Fig. 5). We have also processed the data with Relion and provided information on the workflow (Supplementary Fig. 5).

Minor points:

1) Page 7, line 4 : “hCdt1 that hOrc1-5” should be “hCdt1 and hOrc1-5”

This has been corrected as suggested.

2) Please make sure that Mcm or MCM are homogeneously defined through the manuscript (e.g. page 9 – line 32, Mcm2-7 should be MCM2-7).

Thank you for your comment. This has been corrected as suggested. Please note that MCM2-7 refers to the complex, while Mcm2 refers to the subunits.

3) Page 9-line 32: ... likely due to DNase mediated digestion” this sentence will sound confusing for readers, which may not be familiar with the Dnase1 treatment used in the pre-RC assay. Please revise.

Many thanks. This has been corrected as suggested.

4) The method section does not mention how many times the pull-down experiments shown in figure 1 for example were replicated independently.

Many thanks. This has been corrected as suggested – it was repeated 3 times.

5) Figure 2, panels c and d: the method section should mention how many movies were collected for the two datasets and how many particles were used to determine the percentages DH vs SH.

This has been corrected as suggested. In addition to an updated method section we also provide the image processing workflow in Supplementary Fig. 5.

6) Supplementary figure 7: the figure is nice and informative. However, mapping the conserved residues on a 3D structure will be useful. The Mcm4 sequences diverge but there are also conserved residues. The authors could consider showing a surface representation using consurf?

As suggested a conSURF model of yeast Mcm-4 WHD has been added as panel c to Supplementary Fig. 7. We refrained from a surface representation, as this region was unresolved in the experimental EM density of the hOCCM.

7) Figure 8: number is not correct in the alignment shown panel e (956 not 756 for Mcm6 from *S. cerevisiae*).

Many thanks. This has been corrected as suggested.

Reviewer #2 (Remarks to the Author):

In the present manuscript, Jennifer et al. report the in vitro reconstitution of human DNA replication licensing using purified components (MCM2-7, ORC1-5, ORC6, CDC6 and CDT1). This assay is used to compare human and yeast DNA licensing system. In contrast to yeast, hOrc6 is found to be inessential for recruitment of the first MCM2-7 hexamer while hCdt1 does not show a tight association with hMCM2-7 in solution and is also found to be inessential for the initial recruitment of hMCM2-7. In addition to reconstitution assays, a structure of hOCCM is also solved, providing insights into the overall complex organization.

Overall, the work is of high significance and interest to readers from multiple fields. The field of eukaryotic DNA replication initiation has long focused on yeast as a model system, but recent evidence has indicated that metazoan DNA replication systems may have important differences. The findings systematically recapitulate DNA licensing process with human factors, considerably advancing the field's understanding of this process. Providing the presentation of data can be better organized (especially the somewhat haphazard callout of figure panels) and a few extant questions concerning the approaches resolved, publication would seem warranted.

We thank the reviewer for the kind words and valuable suggestions for further improvement.

Primary comments:

1. Fig. 1c and 1g. It is claimed that there are differences in the binding efficiency between Orc1-5 and DNA +/- Cdc6 and in Mcm2-7 loading efficiency +/- Orc6. The gels suggest this is true, but the differences are too subtle to be sure. Quantitation of the species across replicates would be helpful.

Thank you for the suggestion. We have performed the quantification across three replicates as suggested in Supplementary Fig. 2A and Fig. 4C.

2. On page 7 it's said that "hOrc6 predominantly functions during loading of the second MCM2-7 hexamer", but it's not explicated shown whether hOrc6 is required for this event. Biochemical assays and negative stain EM should be included to answer this question.

We have now provided biochemical data showing that hOrc6 functions post-ATP-hydrolysis in recruiting the 2nd hexamer, and electron microscopy data showing increased MCM2-7 double-hexamer formation in the presence of Orc6 (Fig. 4).

3. Perhaps it was missed during reading, but in Fig. 3c, please explain why the amount of hCdc6 decreases in lane 2 compared to lane 1. Is hCdt1 promoting ATP hydrolysis and hCdc6 release?

Thank you for the insightful question. We agree with the conclusion of the reviewer and added the suggested explanation in the revised manuscript: "The high levels of hCdc6 retention suggests that hCdt1 is necessary for hCdc6 release, which can be observed in the presence of the complete reaction." "However, Cdt1 plays an essential role for ATP-hydrolysis dependent Cdc6 release (Fig. 3a and 4d)."

4. On a related note, why does the binding hOrc6 and hCdc6 to hOrc1-5 seem to be reciprocally antagonistic (e.g., Fig. 1h and 3c)? Is hCdt1 and/or Cdc6 ATPase activity controlling hOrc6 association?

Based on the thoughtful suggestion, we have added a more detailed explanation, highlighting that ATP-hydrolysis controls both steps. However, these events are unlikely due to direct interactions, but rather represent a reorganisation of the overall complex.

5. In previous study (ref. #27), hORC subunits were found to form a complex in an ATP-dependent manner. It may be worth testing whether hOrc6 binds to hOrc1-5 in the presence of ATP at low salt (not 200mM NaCl), since hOrc6 was seen to remain bound to preRC under these conditions in the presence of ATP (Fig. 1h).

We appreciate the suggestion. Indeed, we carried out experiments in the presence of ATP and low salt. We employed potassium glutamate, which has a much lower ionic strength than NaCl. Thus, we conclude that hOrc6 does not form a stable complex with hOrc1-5

6. Please state the protein concentrations used for the pre-RC formation and pull-down experiments.

Many thanks. This has been done as suggested.

7. The binding of hCdt1 to hMcm2-7 is ruled out using mass photometry; however, this was done at very low concentrations of hMcm2-7 (~7.5nM) and hCdt1 (~15nM) and in the absence of ATP. Binding should be examined at a higher protein concentration and in the presence of ATP.

We appreciate the reviewer's comment. We had a similar concern and, therefore, included experiments with a 10-fold excess of hCdt1 and ATP, but we still did not observe hCdt1 binding to hMCM2-7 (Supplementary Fig. 2).

8. Fig. 6d. It is surprising that the ATPase activity of Orc1 is not necessary for Mcm2-7 loading. However, the experiment compares a single WB mutant in Cdc6 to a double WB Orc1 and RF Orc4 mutant. Does the Cdc6 WB mutant behave differently if it is paired with an RF Orc1 mutation?

We appreciate that single and double mutants are not an ideal comparison. As the literature is poor on the impact of double RF/WB mutants, we focussed on the single WB mutants in Cdc6 and Orc1 in the revised Fig. 7. This way one can compare mutants in a more straightforward manner.

9. The build for the DNA looks relatively poor, with a fair degree of backbone and base distortions (as evidenced by the buckling of the cartoon rendering). When building and refining into low resolution maps, it is important that restraints (strong geometry weighting, maintaining appropriate H-bonding distances between base pairs and in secondary structural elements) be applied and enforced. The model should be reexamined and refined accordingly.

We agree with the reviewer and have rebuilt the DNA and updated the figures accordingly. ORC is known to deform the DNA, making the process difficult, but the new DNA model is improved.

10. Fig. 8a. The observation that a different cohort of Mcm WHDs contact ORC-Cdc6 is interesting. Could this be because the Orc1-5/Cdc6 ring is cracked (as shown in the figure) for human but not yeast? And why is the ring cracked in one system but not the other? (It would be helpful to keep the rendering style the same between the two proteins.)

Apologies for the misunderstanding. The crack is in MCM2-7, not ORC-Cdc6. Therefore, we have clarified the figure legend and figure.

11. Given the low resolution of the maps, it is important to show the density associated with all claimed interactions. (e.g., Figs. 6a, Figs. 7e, SF5c, SF6c)

This has been done as suggested in Figs. 7, 8 and 9, Supplementary Figs. 4 and 7.

Minor points:

12. Please provide references for the last couple of sentences in the 2nd paragraph of the Introduction.

Many thanks. This has been changed as suggested.

13. For Fig. 1g, a panel of LS washes should be added as a control.

This has been provided as suggested in the revised Figures 3 and 4.

14. It would be helpful to expand the discussion concerning the role of hCdt1, instead of only one conclusion “Thus, the data suggest that hCdt1 and hMCM2-7 do not form a complex in solution but act synergistically during pre-RC formation to promote hMCM2-7 recruitment.” The structural analysis of hCdt1 is also largely missing, only the interface between hMcm6 and hCdt1 is discussed.

As suggested, we have added a sentence on Cdt1’s connection to ATP-hydrolysis: “Surprisingly, we observed that Cdt1 was necessary for Cdc6 release (Fig. 3c). Considering that Cdc6 release is ATP-hydrolysis dependent (Fig. 1h), the data suggest that Cdt1 is required for induction of ATP-hydrolysis and Cdc6 release”. “However, Cdt1 plays an essential role for ATP-hydrolysis dependent Cdc6 release (Fig. 3a and 4d).” Although the structural analysis of Cdt1 described all three interaction regions with MCM2-7, we have now discussed what structural feature may allow Cdt1 interaction with MCM2-7. “We note that the hMCM2-7 N-terminal domains are rotated relative to the C-terminal domain when comparing hMCM2-7 and hOCCM.”

15. hCdt1 didn’t co-purify with MCM2-7, but hCdt1 is solved in the OCCM complex structure. Does hCdt1 only interact with loaded MCM2-7?

Many thanks for this great question. We discussed this now: “Considering that hCdt1 and hMCM2-7 do not interact in solution, we suggest that initial recruitment of hMCM2-7 to hOrc1-5-hCdc6 leads to a reorganisation of hMCM2-7, which then allows hCdt1 to interact with hMCM2-7. What structural change could be involved in the hMCM2-7 reorganisation? We note that the hMCM2-7 N-terminal domains are rotated relative to the C-terminal domain when comparing hMCM2-735 and hOCCM, which could support an optimal hCdt1 interaction surface.”

16. In the second paragraph of section “hOrc1-5 binding to DNA is stabilised by hCdc6”, it is stated that “While hOrc1-5 only associated with dsDNA alone and only background binding of hOrc6 was

observed (compare lane 4 with lanes 2 and 3)" The figure panel showing these lanes should be noted in the text.

Many thanks. This has been changed as suggested.

17. The callout of Figure 5 in section "hMCM2-7 encircles DNA and adopts an open ring structure" seems to have some inconsistencies. For example, Fig. 5d doesn't seem to show that "DNA is not visible within the N-terminal Mcm2-7 ring section". And from Fig. 5c it cannot be concluded that the "hMcm2-7 ring is partially opened at the Mcm2-Mcm5 gate". Fig. 5d also doesn't show yCdt1 for comparison with hCdt1.

We apologise for referencing the wrong figure. This has been corrected.

18. The title of Fig. 6 seems wrong? It's not Cdc6 that is restructured but hOrc1?

Many thanks. We revised the title as suggested: "The hOrc1-hCdc6 interface becomes restructured in the hOCCM and hCdc6 ATPase is crucial for complex disassembly."

19. For the statement, "In contrast, in hOrc2-530 the hOrc2-WHD was folded inwards blocking DNA interactions (Fig. 9a, centre), something also seen in the Drosophila Orc1-6 complex", a citation is needed for the final clause.

Many thanks. This has been changed as suggested.

20. Fig. 5a. Please add a color key for identifying subunits.

Many thanks. This has been changed as suggested.

21. In figure 7c, hOrc2 loop starts and ends with the same residue number (364).

Many thanks. This has been changed as suggested.

22. On page 15, "Unfortunately, purification of this mutant MCM2-7 complex resulted in poor Mcm5 retention (Fig. 8)". Here the Fig. 8 should be Fig. 8d.

Many thanks. This has been changed as suggested.

23. On page 15, it is stated "Consistent with this, we observed that the hMcm6-WHD is similarly organised as the yMcm6 WHD. It is sandwiched between Orc4, Orc5 and Cdt1 (Fig. 5c, d, Supplementary Fig 5b)." Fig. 5c,d doesn't show hMcm6-WHD.

Many thanks. This has been changed as suggested.

24. On page 15, it is stated "Here, we observed a close contact between hCdt1 (aa413-424) and hMcm6 (aa756-786)". Should '756-786' be '756-768'?

Many thanks. This has been changed as suggested.

25. At the end of page 16, “Consistent with the essential role of Mcm3 in DNA replication and cellular viability, the impact of the mutations was not severe.” Why is Mcm3 essential, but the mutations don’t have severe consequences?

Thank you for the comment. We revised the section “Although hMCM2-7loading was impacted by the hCdc6 mutation, the cancer cells were growing. This could be because most cancer-associated mutations are heterozygous and, thus, mutations can be tolerated. However, reduced hMCM2-7loading would render these cells more prone to genome stability and sensitise them to chemotherapy.”

26. Fig. SF1d. Why are there multiple masses for MCM2-7?

Human MCM2-7 tends to form multiple complex species, particularly at the low protein concentration used in the mass-photometry measurements. We clarified the figure legend accordingly.

27. In Fig. S4, the iterative Homo/Hetero Refinement and Subset Selection step is confusing. Please simplify this.

Thank you for the comment. Accordingly, the figure has been clarified.

28. It would be appropriate to mention the other two preprints posted online recently that are pertinent to the work:

(<https://www.biorxiv.org/content/10.1101/2024.04.10.588796v1>,
<https://www.biorxiv.org/content/10.1101/2024.04.10.588848v1>)

At the time point of submission, these were not online, but we have added them now.